# Patched 1 reduces the accessibility of cholesterol in the outer leaflet of membranes

**Maia Kinnebrew[1], Giovanni Luchetti[1,2], Ria Sircar[1], Sara Frigui[1], Lucrezia Vittoria Viti[3], Tomoki Naito[4], Francis Beckert[1], Yasunori Saheki[4], Christian Siebold[3], Arun Radhakrishnan[5], Rajat Rohatgi[1]***

[1]Department of Biochemistry and Medicine, Stanford University School of Medicine, Stanford, United States; [2]Department of Physiological Chemistry, Genentech, South San Francisco, United States; [3]Division of Structural Biology, Wellcome Centre for Human Genetics, University of Oxford, Oxford, United Kingdom; [4]Lee Kong Chian School of Medicine, Nanyang Technological University, Singapore, Singapore; [5]Department of Molecular Genetics, University of Texas Southwestern Medical Center, Dallas, United States

**Abstract** A long-standing mystery in vertebrate Hedgehog signaling is how Patched 1 (PTCH1), the receptor for Hedgehog ligands, inhibits the activity of Smoothened, the protein that transmits the signal across the membrane. We previously proposed (Kinnebrew et al., 2019) that PTCH1 inhibits Smoothened by depleting accessible cholesterol from the ciliary membrane. Using a new imaging-based assay to directly measure the transport activity of PTCH1, we find that PTCH1 depletes accessible cholesterol from the outer leaflet of the plasma membrane. This transport activity is terminated by binding of Hedgehog ligands to PTCH1 or by dissipation of the transmembrane potassium gradient. These results point to the unexpected model that PTCH1 moves cholesterol from the outer to the inner leaflet of the membrane in exchange for potassium ion export in the opposite direction. Our study provides a plausible solution for how PTCH1 inhibits SMO by changing the organization of cholesterol in membranes and establishes a general framework for studying how proteins change cholesterol accessibility to regulate membrane-dependent processes in cells.

**\*For correspondence:**
rrohatgi@stanford.edu

## Editor's evaluation

This paper addresses an important question regarding the mechanisms of Hedgehog signaling. The authors develop a new method to observe changes in cholesterol accessibility in the outer lamella of the plasma membrane to investigate the activity of the Hedgehog receptor PTCH1 and its modulation by Sonic Hedgehog. The results support the conclusion that PTCH1 needs a potassium gradient to reduce chemically-active cholesterol in the outer lamella, presumably by translocating the sterol to the inner lamella. The proposed model contradicts previous reports that suggest transport in the opposite direction using the plasma membrane sodium gradient for energy. In the initial review, the reviewers appreciated the potential impact of the findings and suggested several areas for improvement. The authors have now satisfactorily addressed the reviewers' comments in the revised manuscript.

## Introduction

Patched 1 (PTCH1) is a 12-pass transmembrane (TM) receptor for Hedgehog (Hh) ligands that was cloned over 30 years ago in *Drosophila* (*Hooper and Scott, 1989*; *Nakano et al., 1989*). PTCH1 keeps Hh signaling in the off state by inhibiting the function of Smoothened (SMO), a member of the

Frizzled-class G protein-coupled receptor family (reviewed in *Kong et al., 2019*). Hh ligands like Sonic Hedgehog (SHH) bind and inactivate PTCH1, allowing SMO to adopt an active conformation and transduce the Hh signal across the membrane. Inactivating mutations in PTCH1 lead to unrestrained Hh signaling and cause birth defects and human cancers. Despite three decades of research, the biochemical activity of PTCH1, and how this activity inhibits SMO, has remained mysterious.

An early clue to the function of PTCH1 (confirmed more recently by structural studies) was the observation that its sequence was similar to Niemann-Pick C1 (NPC1), a protein that transports cholesterol from the lumen of the lysosome to the cytoplasm (*Carstea et al., 1997*; *Gong et al., 2018*; *Gong et al., 2016*; *Li et al., 2016*; *Loftus et al., 1997*; *Qian et al., 2020*; *Qi et al., 2018a*; *Qi et al., 2018b*; *Zhang et al., 2018*). PTCH1 overexpression was also shown to enhance the efflux of a fluorescent analog of cholesterol (BODIPY-cholesterol) from cells (*Bidet et al., 2011*). A second clue was the observation that both oxysterols (*Nachtergaele et al., 2012*) and cholesterol (*Byrne et al., 2016*; *Huang et al., 2016*; *Luchetti et al., 2016*) directly bind SMO and were sufficient to activate Hh signaling (even in the absence of Hh ligands). These latter studies led to the model that PTCH1 uses its transporter function to somehow prevent cholesterol from binding and activating SMO (*Kong et al., 2019*; *Kowatsch et al., 2019*).

To identify the second messenger that communicates the signal between PTCH1 and SMO, we previously conducted an unbiased CRISPR screen to find cellular lipids that influence Hh signaling (*Kinnebrew et al., 2019*). The screen identified multiple genes encoding enzymes at all levels in the cholesterol biosynthesis pathway as positive regulators, confirming the central importance of cholesterol in Hh signaling. More importantly, it implicated a minor pool of membrane cholesterol, termed accessible cholesterol, in the communication between PTCH1 and SMO. A large body of data (summarized in *Das et al., 2014*; *Lange and Steck, 2020*; *McConnell and Radhakrishnan, 2003*) supports the model that cholesterol in the plasma membrane segregates into at least two pools. The major pool of plasma membrane cholesterol is sequestered in complexes with phospholipids, particularly sphingomyelin (SM), and is inaccessible to proteins or soluble acceptors. In contrast, a minor pool of plasma membrane cholesterol is free from interactions with other lipids and is accessible to modulate the activity of proteins or to escape the membrane to soluble acceptors. The major physico-chemical difference between cholesterol in these two pools is its chemical activity (a quantity related to chemical potential), which is reduced by the formation of complexes with membrane phospholipids. Selectively changing accessible cholesterol (without altering total cholesterol) in membranes altered Hh signaling strength in a predictable manner: increasing accessible cholesterol by depleting SM potentiated Hh signaling while trapping accessible cholesterol using a bacterial toxin domain (ALOD4) attenuated Hh signaling (*Kinnebrew et al., 2019*; *Radhakrishnan et al., 2020*).

These studies led us to propose the model that PTCH1 depletes accessible cholesterol to block SMO activity (*Kinnebrew et al., 2019*; *Radhakrishnan et al., 2020*). To test this hypothesis, we established a direct assay for the biochemical activity of PTCH1 and used it to investigate the transport mechanism.

## Results

### Conflicting models for the influence of PTCH1 on membrane cholesterol organization

Previous studies have evaluated the effect of PTCH1 on membrane cholesterol by measuring the steady-state binding of fluorescent cholesterol sensors derived from microbial toxins like Perfringolysin O (PFO). Using a fluorescently labeled, non-lytic version of PFO called PFO*, we found that PTCH1 inactivation by SHH leads to increased accessible cholesterol in the *outer leaflet* of the membrane surrounding primary cilia, the cellular compartment where SMO triggers downstream signaling (*Kinnebrew et al., 2019*). Studies from other groups using PTCH1 expressed throughout the plasma membrane concluded that PTCH1 inactivation increases cholesterol levels in the *inner leaflet* (*Zhang et al., 2018*). This study used a probe derived from the cholesterol-binding D4 domain of PFO (PFOD4) carrying several mutations and modifications that increased its cholesterol sensitivity (*Liu et al., 2017*). One important difference between the two studies is that the engineered PFOD4 probes used in the latter study are not selective for accessible cholesterol, binding to cholesterol regardless of the surrounding phospholipid environment (*Liu et al., 2017*).

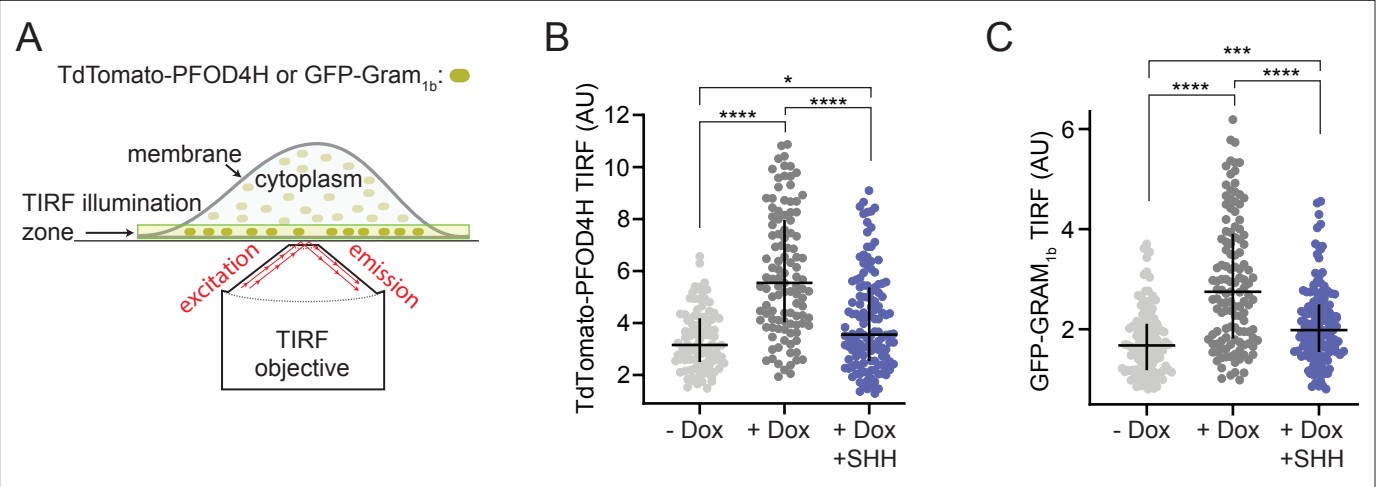

**Figure 1.** PTCH1 activity increases inner leaflet cholesterol. (**A**) Total internal reflection fluorescence microscopy (TIRFM) was used to measure the recruitment of two different fluorescent cholesterol sensors (TdTomato-PFOD4H or GFP-GRAM$_{1b}$) to the inner leaflet of the basal plasma membrane adjoining the coverslip. TIRFM only excites the sensor molecules (dark green) present in the ~100-nm-wide illumination zone, thus excluding fluorescence from the majority of sensor molecules present in the cytoplasm or bound to internal organelles (light green). (**B, C**) Steady-state TdTomato-PFOD4H (**B**) or GFP-GRAM$_{1b}$ (**C**) fluorescence at the inner leaflet of the plasma membrane in HEK293T cells with or without PTCH1 expression induced by the addition of Doxycycline (Dox). SHH (1 μM) was used to inactivate PTCH1 where indicated. Each circle represents the average intensity of fluorescence from a single cell membrane (n>65 cells), with the horizontal line representing the median and the vertical line denoting the interquartile range. Statistical significance was determined with a Mann-Whitney test. p-values in (**B**) are: −Dox vs. +Dox<0.0001, +Dox vs. +Dox + SHH<0.0001 and −Dox vs. +Dox + SHH=0.0127. p-values in (**C**) are: −Dox vs. +Dox<0.0001, +Dox vs. +Dox + SHH<0.0001 and −Dox vs. +Dox + SHH=0.0002. Experiments in (**B**) and (**C**) were repeated three independent times with similar results.

The online version of this article includes the following figure supplement(s) for figure 1:

**Figure supplement 1.** PTCH1 activity decreases outer leaflet cholesterol.

To measure inner leaflet accessible cholesterol in membranes of live cells with or without PTCH1, we used a version of PFOD4 containing a single point mutation (D434S), called PFOD4H, that retains selectivity for accessible cholesterol (*Abe and Kobayashi, 2021*; *Gay et al., 2015*; *Johnson et al., 2012*; *Maekawa and Fairn, 2015*). Membrane recruitment of cytoplasmically expressed PFOD4H has been shown to directly reflect the abundance of inner leaflet cholesterol (*Maekawa and Fairn, 2015*). Using a plasmid that fused the coding sequence of PFOD4H to the TdTomato fluorescent protein, we expressed TdTomato-PFOD4H in the cytoplasm of HEK293T cells and monitored its recruitment to the inner leaflet of the plasma membrane by total internal reflection fluorescence microscopy (TIRFM) (*Maekawa and Fairn, 2015*). The limited axial depth of fluorophore excitation by TIRFM allowed us to readily measure the amount of TdTomato-PFOD4H bound to the plasma membrane in live cells, without interference from the TdTomato-PFOD4H in the cytoplasm or bound to intracellular organelles (*Figure 1A*). To evaluate the effect of PTCH1, we used HEK293T cells stably expressing PTCH1 under the control of a Doxycycline (Dox)-inducible promoter. PTCH1 expression increased TdTomato-PFOD4H binding to the inner leaflet of the plasma membrane in a manner that could be blocked by its inhibitory ligand SHH (*Figure 1B*), consistent with the model that PTCH1 increases inner leaflet cholesterol accessibility.

Since these results were different from those reported previously (*Zhang et al., 2018*), we repeated these measurements using a eukaryotic cholesterol-binding domain, known as a GRAM domain, from the GRAMD1b protein fused to GFP (GFP-GRAM$_{1b}$). GFP-tagged GRAM domains expressed in cells have recently been used to measure changes in inner leaflet cholesterol in response to cholesterol loading or extraction using TIRFM (*Ercan et al., 2021*; *Naito et al., 2019*). GRAM domains bind to excess accessible cholesterol in the inner leaflet of the plasma membrane in conjunction with phosphatidylserine, allowing the GRAMD1 proteins to transport accessible cholesterol from the plasma membrane to the endoplasmic reticulum (ER) at ER-plasma membrane contact sites (*Ercan et al., 2021*; *Ferrari et al., 2020*; *Naito et al., 2019*; *Naito and Saheki, 2021*; *Sandhu et al., 2018*). PFOD4H and GRAM$_{1b}$ have no structural similarity to each other, with the former derived from a bacterial toxin

and the latter from a eukaryotic protein. Steady-state binding measurements of GFP-GRAM$_{1b}$ again supported the model that PTCH1 increases inner leaflet cholesterol accessibility (*Figure 1C*).

To measure outer leaflet cholesterol accessibility in these same cells, we used a fluorescently labeled cholesterol-binding domain (mNeon-ALOD4) from a different microbial toxin called Anthrolysin O (ALO) (*Endapally et al., 2019*; *Johnson and Radhakrishnan, 2021*). We observed small but consistent changes in outer leaflet cholesterol that were complementary to those seen at the inner leaflet: PTCH1 activity decreased outer leaflet cholesterol accessibility (*Figure 1—figure supplement 1*). These results are in agreement with our previous observations that PTCH1 decreases cholesterol accessibility in the outer leaflet of the ciliary membrane of fixed cells (*Kinnebrew et al., 2019*).

Given the apparently conflicting results using steady-state binding of probes based on pore-forming toxins like PFO or ALO and the caveats associated with these probes (noted previously by independent investigators [*Courtney et al., 2018*]), we sought to measure the effect of PTCH1 on membrane cholesterol by developing a completely different assay.

## A kinetic assay for accessible cholesterol in membranes

To test the hypothesis that PTCH1 can reduce accessible cholesterol in membranes of live cells, we first established an assay to measure this pool of cholesterol. Methyl-β-cyclodextrin (MβCD) is a cyclic oligosaccharide with a polar surface and a hydrophobic cavity that can rapidly extract cholesterol from both model and cellular membranes (*Christian et al., 1997*; *Zidovetzki and Levitan, 2007*). MβCD is a useful probe for the organization of cholesterol in the membrane because it bypasses the rate-limiting step in cholesterol extraction, the energetically unfavorable transfer of cholesterol from the membrane to the adjacent aqueous layer (*Kilsdonk et al., 1995*). MβCD is also non-intrusive because it does not destabilize lipid-lipid interactions or change membrane surface properties (*Ohvo and Slotte, 1996*).

A large body of experimental and theoretical work on defined lipid monolayers and bilayers has shown that the rate constant for cholesterol removal by MβCD is proportional to the chemical activity or accessibility of cholesterol (*Lange et al., 2004*; *Litz et al., 2016*; *McConnell and Radhakrishnan, 2003*; *Ohvo and Slotte, 1996*; *Radhakrishnan and McConnell, 2000*). For instance, SM, the phospholipid that forms the most stable complexes with cholesterol, markedly reduces the rate of cholesterol removal by MβCD from lipid monolayers (*Ohvo and Slotte, 1996*). MβCD (which is membrane impermeable) also extracts cholesterol from intact cells when added to the extracellular medium by accepting cholesterol that desorbs from the outer leaflet of the plasma membrane. Interestingly, MβCD-catalyzed cholesterol efflux from cells follows biexponential kinetics, suggesting the presence of two kinetic pools of cholesterol: a fast pool with a $t_{1/2}$ of <1 min and a slow pool with a $t_{1/2}$ of >10 min (*Yancey et al., 1996*). The fast pool represents the accessible cholesterol pool because its size can be expanded by depleting SM and other phospholipids that sequester cholesterol (*Haynes et al., 2000*).

Since cholesterol can exchange rapidly between the two leaflets of the plasma membrane, removal of cholesterol from the outer leaflet with MβCD should also deplete cholesterol from the inner leaflet (*Figure 2A*). Consequently, we reasoned that the decrease in inner leaflet cholesterol could be used to monitor the extraction of cholesterol from the outer leaflet by MβCD. To measure changes in inner leaflet cholesterol of live cells, we used TIRFM to measure membrane recruitment of cytoplasmically expressed TdTomato-PFOD4H (*Figure 2A, B*). Addition of MβCD to the extracellular medium led to a decrease in TdTomato-PFOD4H fluorescence at the plasma membrane and, concomitantly, an increase in TdTomato-PFOD4H fluorescence in the cytoplasm, consistent with recent observations in HeLa cells (*Abe and Kobayashi, 2021*; *Figure 2C*).

Instead of using the absolute steady-state values of TdTomato-PFOD4H fluorescence at the basal membrane, we used the kinetics of extraction as a measure of cholesterol accessibility. Kinetic assays are generally more sensitive and less susceptible to artifacts. For example, the steady-state binding of TdTomato-PFOD4H to the inner leaflet can be influenced by many factors other than the cholesterol content, such as the amount of TdTomato-PFOD4H expressed in an individual cell and the non-specific affinity of the probe for membranes (*Courtney et al., 2018*). In contrast, cholesterol extraction by MβCD follows first-order, exponential decay kinetics (*Figure 2D*): the half-life ($t_{1/2}$) for extraction is independent of the starting baseline value (as it is for radioactive decay). To reliably measure the kinetics of cholesterol extraction under each condition, we combined results from multiple individual cells. Each of these cells shows a different baseline level of TdTomato-PFOD4 fluorescence at the

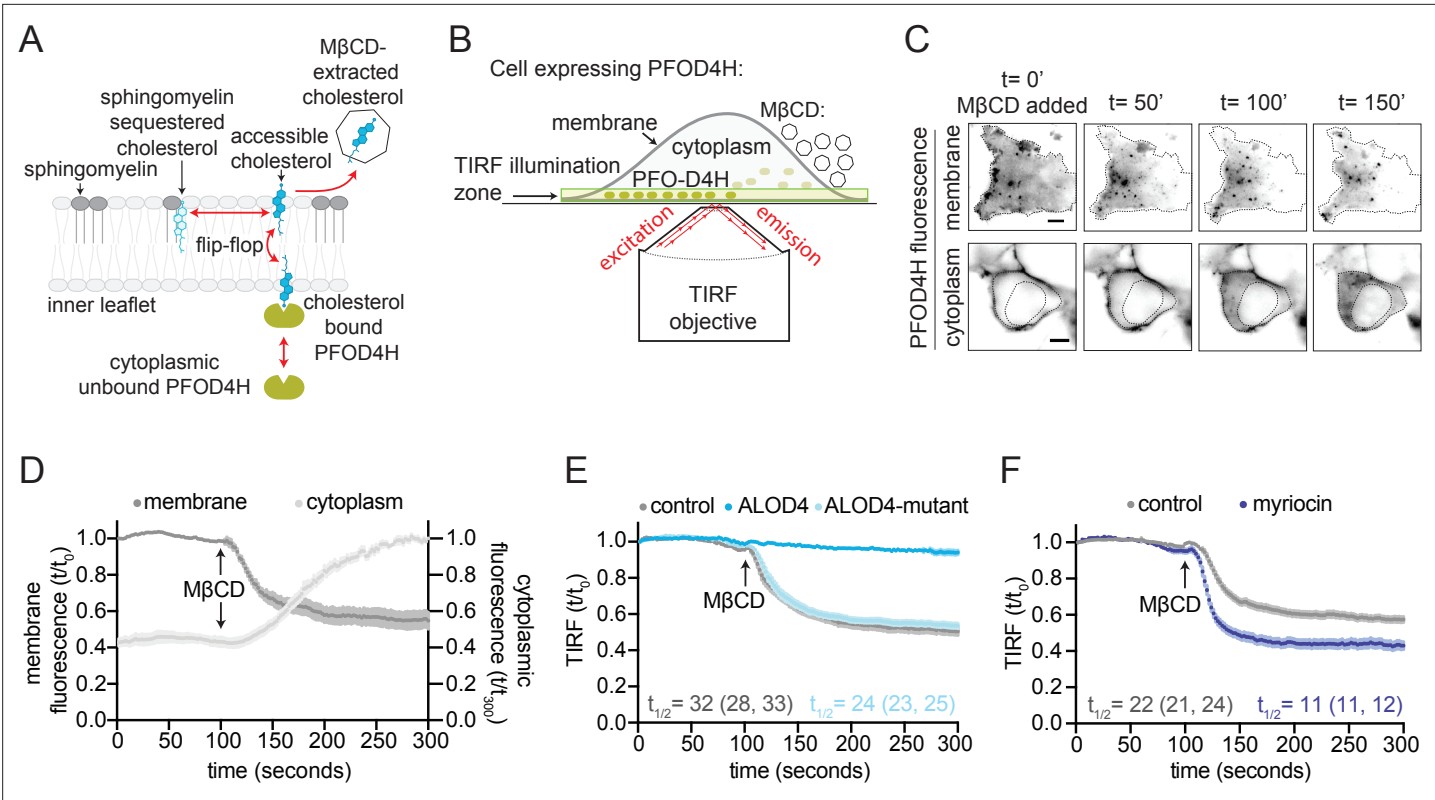

**Figure 2.** Measurement of outer leaflet accessible cholesterol in live cells with total internal reflection fluorescence microscopy (TIRFM). (**A**) Extraction of outer leaflet accessible cholesterol by MβCD results in detachment of TdTomato-PFOD4H from the inner leaflet of the plasma membrane due to flip-flop of cholesterol between the two leaflets. Outer leaflet cholesterol can exchange between two pools: accessible cholesterol (dark blue) or sequestered cholesterol (light blue). Sphingomyelin (dark gray), present exclusively in the outer leaflet, plays a dominant role in sequestering cholesterol. Accessible cholesterol (drawn as projecting further out of the surface of the membrane into the aqueous boundary layer) has higher chemical activity and thus greater propensity to interact with proteins or to be transferred to a soluble acceptor like MβCD. (**B**) Schematic of the assay to detect outer leaflet accessible cholesterol using TIRFM. TdTomato-PFOD4H (green ovals) expressed intracellularly in HEK293T cells is recruited to the inner leaflet of the basal plasma membrane by binding to cholesterol (see (**A**)), bringing it into the TIRF illumination zone. MβCD added to the media (right) extracts outer leaflet accessible cholesterol, resulting in depletion of inner leaflet cholesterol and detachment of TdTomato-PFOD4H from the membrane. (**C**) Images of TdTomato-PFOD4H at the membrane (top panel, TIRFM) or in the cytoplasm (bottom panel, epifluorescence microscopy) of a single cell after MβCD addition (316 μM). Dotted lines mark the cell border (top row) or nuclear border (bottom row). Scale bar is 5 μm. (**D**) Time course of the change in TdTomato-PFOD4H membrane and cytoplasmic fluorescence (measured every 2 s by TIRFM or epifluorescence microscopy, respectively) after MβCD addition (316 μM) (quantified from images of the type shown in (**C**)). Black arrows mark the time point in the experiment when MβCD was added. Each curve represents the mean fluorescence from six or more cells, with standard error of the mean (SEM) shown in lighter shading around each curve. Fluorescence is depicted relative to the starting fluorescence ($t/t_0$) for the TIRFM data (left y-axis) and relative to the ending fluorescence ($t/t_{300}$) for the epifluorescence microscopy data (right y-axis). This experiment was repeated five independent times with similar results. (**E, F**) Time course of the change in TdTomato-PFOD4H membrane fluorescence measured by TIRFM after MβCD addition (316 μM, arrow) in control cells or in cells subjected to two treatments. In (**E**), cells were pre-incubated with ALOD4 or an ALOD4 mutant defective in cholesterol binding (added at 5 μM for 45 min prior to the assay). In (**F**), cells were treated with 80 μM myriocin for 3 days to deplete cellular SM. Each curve shows the mean fluorescence measured from >20 cells taken from at least three biological replicates, with SEM depicted in lighter shading around each curve. Fluorescence is depicted relative to the starting fluorescence ($t/t_0$) for the TIRFM data. Curves showing data without normalization to baseline fluorescence values are provided in *Figure 2—figure supplement 1C, D*. In (**E**) and (**F**), curve fits (see Materials and methods) were used to determine the time required for TdTomato-PFOD4H fluorescence at the plasma membrane to drop by one-half of its starting value ($t_{1/2}$), with the 95% confidence interval (CI) shown in parentheses. The experiment was repeated three independent times with similar results.

The online version of this article includes the following figure supplement(s) for figure 2:

**Figure supplement 1.** Characterization of a kinetic assay to measure outer leaflet accessible cholesterol.

membrane, likely because each cell expresses different amounts of the probe (see the wide distribution of baseline values in *Figure 1B*). To focus on the kinetics (or $t_{1/2}$ of extraction) we normalized the fluorescence for each cell to its own baseline value (at time=0) before averaging over all the cells analyzed. Extraction curves without baseline normalization are shown in the figure supplements for selected panels in *Figures 2 and 3*.

After addition of MβCD, the $t_{1/2}$ of TdTomato-PFOD4H detachment was <1 min (*Figure 2D*), consistent with the rapid kinetics of accessible cholesterol depletion by MβCD observed in prior studies using both defined lipid monolayers and cells (*Haynes et al., 2000*; *Ohvo and Slotte, 1996*; *Radhakrishnan and McConnell, 2000*; *Yancey et al., 1996*). The detachment of TdTomato-PFOD4H from the inner leaflet of the plasma membrane and its release into the cytoplasm indicates that removal of cholesterol by MβCD from the outer leaflet also causes the depletion of cholesterol from the inner leaflet, confirming the rapid exchange or flip-flop of cholesterol between the two leaflets (*Figure 2A*).

To test if the kinetics of MβCD-induced TdTomato-PFOD4H detachment was influenced by the size of the outer leaflet accessible cholesterol pool, we increased or decreased membrane accessible cholesterol using two chemically distinct perturbations. First, the cholesterol-binding D4 domain of the bacterial toxin Anthrolysin O (ALOD4) was added to the extracellular medium to trap outer leaflet accessible cholesterol. This protein binds specifically to accessible cholesterol and sequesters it without altering the total cholesterol levels in the membrane (*Infante and Radhakrishnan, 2017*). Consistent with its cholesterol-sequestering properties, wild-type (WT) ALOD4 (but not a mutant that cannot bind cholesterol) nearly abolished MβCD-induced detachment of TdTomato-PFOD4H from the membrane (*Figure 2E*). We observed that ALOD4 caused a decrease in the steady-state TdTomato-PFOD4H binding to the inner leaflet, likely because sequestration of accessible cholesterol in the outer leaflet drove the movement of cholesterol from the inner to the outer leaflet by mass action (*Figure 2—figure supplement 1A*).

To increase the pool of outer leaflet accessible cholesterol, we depleted cells of SM, a lipid exclusively present in the outer leaflet of the plasma membrane that plays a dominant role in sequestering accessible cholesterol. SM depletion liberates accessible cholesterol without altering total cholesterol in the plasma membrane (*Das et al., 2014*; *Tafesse et al., 2015*). Myriocin, a drug commonly used to deplete SM, increased the steady-state binding of TdTomato-PFOD4H to the inner leaflet, likely because some of the accessible cholesterol liberated from the outer leaflet moved to the inner leaflet (again showing free exchange of accessible cholesterol between the two leaflets) (*Figure 2—figure supplement 1B*). Depletion of SM both accelerated the rate of cholesterol extraction by MβCD and increased the total amount of cholesterol extracted, consistent with expansion of the outer leaflet pool of accessible cholesterol (*Figure 2F*). Taken together, our TIRFM assay shows that the kinetics of MβCD-induced detachment of TdTomato-PFOD4H in live cells can be used to measure the pool of accessible cholesterol in the outer leaflet of the plasma membrane. Although TdTomato-PFOD4H is bound to the inner leaflet and MβCD extracts cholesterol from the outer leaflet (*Figure 2A*), the rapid exchange of cholesterol between the two leaflets allows detachment of the TdTomato-PFOD4H from the inner leaflet to reflect cholesterol extraction from the outer leaflet.

## PTCH1 reduces accessible cholesterol in the membrane outer leaflet

Armed with this cholesterol transport assay, we directly assessed the effects of PTCH1 on accessible cholesterol in HEK293T cells expressing PTCH1 under the control of a Dox-inducible promoter. Control cell lines expressed a widely used truncation mutant of PTCH1 that lacks its second extracellular loop (hereafter called PTCH1-ΔL2) under control of the same Dox-inducible promoter (*Figure 3A and B*). PTCH1-ΔL2 inhibits SMO activity but cannot bind SHH and thus cannot be inactivated (*Briscoe et al., 2001*; *Marigo et al., 1996*).

Expression of PTCH1 induced by Dox addition prevented MβCD-induced detachment of TdTomato-PFOD4H from the inner leaflet (*Figure 3C*). PTCH1 induction did not change the abundance of TdTomato-PFOD4H in cells (*Figure 3—figure supplement 1A*). Measuring extraction kinetics at a series of MβCD concentrations provided a quantitative assessment of the effect of PTCH1. Without PTCH1 induction, MβCD concentrations as low as 80 μM led to TdTomato-PFOD4H detachment from the membrane; in contrast, in the presence of PTCH1, cells were resistant to MβCD even when concentrations exceeded 300 μM (*Figure 3D, E*). The concentration of MβCD required to achieve the half-maximum detachment rate was increased from ~170 μM to ~700 μM when PTCH1 was expressed

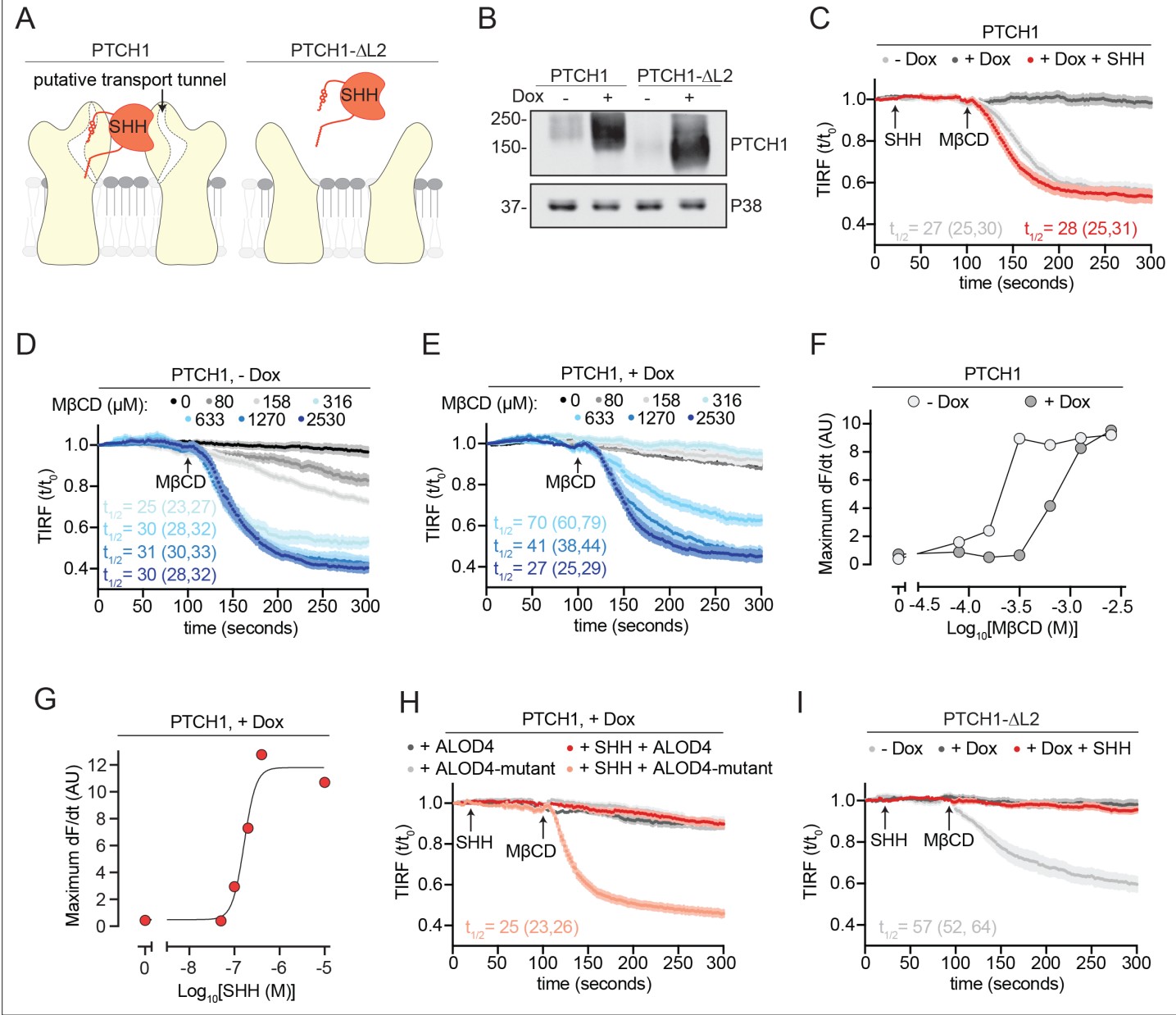

**Figure 3.** PTCH1 depletes outer leaflet accessible cholesterol. (**A**) Cartoon representations of PTCH1 bound to SHH and PTCH1-ΔL2. Deletion of the second extracellular loop (L2) in PTCH1-ΔL2 abolishes both SHH binding and a putative tunnel through PTCH1 that has been proposed to form a conduit for cholesterol transport. (**B**) Immunoblot showing Doxycycline (Dox)-inducible expression of stably expressed PTCH1 and PTCH1-ΔL2 in HEK293T cells. (**C**) Time course of the change in TdTomato-PFOD4H membrane fluorescence after MβCD addition (316 μM, arrow) in the presence (+Dox) or absence (−Dox) of PTCH1 expression. SHH is added (1 μM, red curve only) during imaging to inactivate PTCH1. (**D, E**) The experiment in (**C**) was performed at various concentrations of MβCD either without (**D**, −Dox) or with (**E**, +Dox) PTCH1 expression. (**F**) The maximum rate of change in TdTomato-PFOD4H membrane fluorescence (dF/dt) is depicted as a function of MβCD concentration. At each MβCD concentration, the max dF/dt was calculated from the curves shown in (**D**) and (**E**). (**G**) The maximum rate of cholesterol extraction (dF/dt) by MβCD (316 μM) was measured at increasing concentrations of SHH in cells expressing PTCH1. The individual curves used to measure the max dF/dt values are shown in *Figure 3— figure supplement 1B*. (**H**) Time course of the change in TdTomato-PFOD4H membrane fluorescence after MβCD addition (316 μM, arrow) in PTCH1-expressing cells (+Dox) with or without SHH addition (1 μM, red and pink curves only). Cells were treated with either ALOD4 or an ALOD4 mutant defective in cholesterol binding (added at 5 μM for 45 min prior to assay). (**I**) Time course of the change in TdTomato-PFOD4H membrane fluorescence after MβCD addition (316 μM, arrow) in the presence (+Dox) or absence (−Dox) of PTCH1-ΔL2 expression. Red curve depicts the effect of SHH addition (1 μM, arrow) in PTCH1-ΔL2 expressing cells. For (**C–E**), (**H**), and (**I**), each curve shows the mean fluorescence measured from >20 cells taken from at least three biological replicates, with SEM depicted in lighter shading around each curve. Fluorescence is depicted relative to the starting fluorescence (t/t₀) for the TIRFM data. For (**C**) and (**I**), curves showing data without normalization to baseline fluorescence values are provided in *Figure 3—figure*

*Figure 3 continued on next page*

*Figure 3 continued*

*supplement 2A B*. The $t_{1/2}$, along with the 95% CI, is shown for each curve. All experiments were repeated three independent times, with the exception of (**D**) and (**E**) which were performed two times. CI, confidence interval; SEM, standard error of the mean.

The online version of this article includes the following source data and figure supplement(s) for figure 3:

**Source data 1.** Dotted lines mark the cropped region of the immunoblot that was used to generate panel *Figure 3B*.

**Figure supplement 1.** Effect of PTCH1-ΔL2 on membrane cholesterol accessibility.

**Figure supplement 1—source data 1.** Uncropped original scans of films used for immunoblots shown in *Figure 3—figure supplement 1A*.

**Figure supplement 2.** PTCH1 and PTCH1-ΔL2 reduce outer leaflet cholesterol accessibility.

in the plasma membrane (*Figure 3F*). PTCH1 expression does not completely prevent MβCD from removing membrane cholesterol; it simply shifts the MβCD concentrations required to higher values. Since cyclodextrins like MβCD can also extract the SM-sequestered pool of cholesterol at higher concentrations (*Das et al., 2014*), the most parsimonious interpretation of these data is that PTCH1 reduces the pool of accessible cholesterol in the membrane outer leaflet. Indeed, the effect of PTCH1 mimics the effect of ALOD4 (*Figure 2E*), a protein known to trap accessible cholesterol (*Infante and Radhakrishnan, 2017*).

Two different sets of experiments were used to establish that the ability of PTCH1 to antagonize extraction of membrane cholesterol by MβCD was relevant to its biochemical function in Hh signaling (rather than due to an unrelated or off-target effect of PTCH1 overexpression in membranes). First, the inhibitory effect of PTCH1 on MβCD-induced cholesterol extraction could be reversed by the acute administration of SHH in a dose-dependent manner (*Figure 3C and G*). SHH addition liberated outer leaflet accessible cholesterol in our assay because (1) it restored rapid kinetics of cholesterol extraction by MβCD and (2) its effect could be antagonized by WT ALOD4 (but not a mutant that cannot bind cholesterol) (*Figure 3C and H*). Second, to establish the specificity of SHH, we turned to PTCH1-ΔL2, a truncation mutant that cannot bind to Hh ligands (*Figure 3A and B*). This widely used mutant can act in a dominant fashion to block Hh signaling, even in the presence of Hh ligands (*Briscoe et al., 2001*). Like WT PTCH1, PTCH1-ΔL2 also antagonized MβCD-induced cholesterol extraction but, crucially, this effect could not be reversed by SHH (*Figure 3I*). Interestingly, dose-response analysis demonstrated that PTCH1-ΔL2 appeared to be more effective than WT PTCH1 in reducing outer leaflet accessible cholesterol (*Figure 3—figure supplement 1F*). This difference is

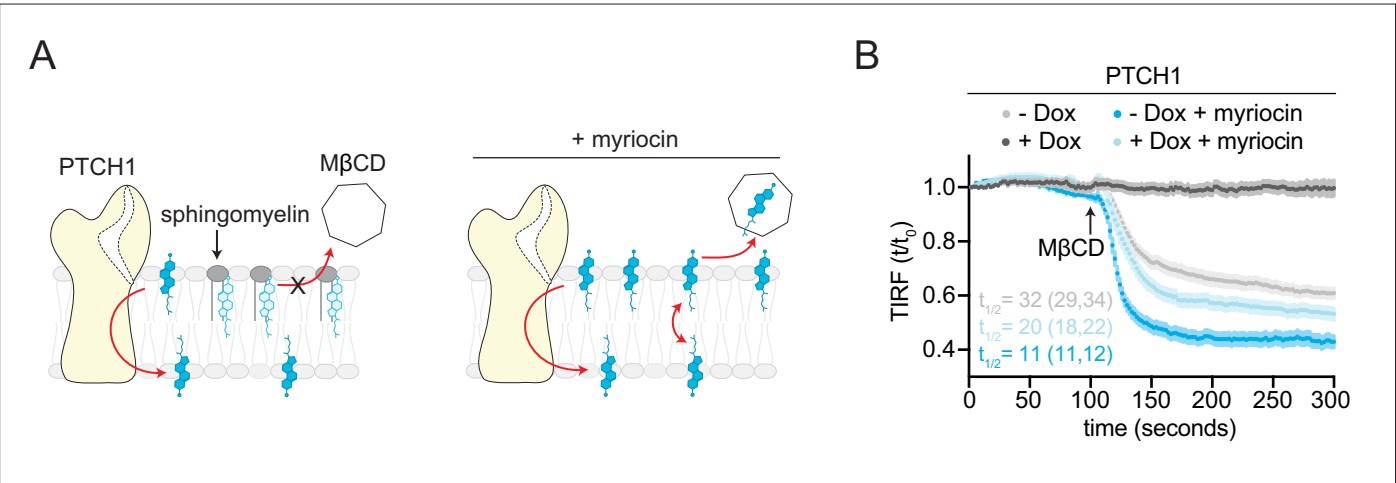

**Figure 4.** Opposing effects of PTCH1 and sphingomyelin (SM) depletion on outer leaflet accessible cholesterol. (**A**) Cartoon showing that depletion of SM with myriocin increases accessible cholesterol in the membrane outer leaflet, increasing the transport load on PTCH1 and preventing it from depleting accessible cholesterol from the outer leaflet. (**B**) Time course of the change in TdTomato-PFOD4H membrane fluorescence after MβCD addition (316 µM, arrow) in the presence (+Dox) or absence (−Dox) of PTCH1 expression and with or without myriocin treatment (80 µM, 3 days) to deplete SM. Each curve shows the mean fluorescence measured from >15 cells taken from at least three biological replicates, with SEM depicted in lighter shading around each curve. Fluorescence is depicted relative to the starting fluorescence (t/t_0) for the TIRFM data. The $t_{1/2}$, along with the 95% CI, is shown for each curve. The experiment was repeated three independent times with similar results. CI, confidence interval; SEM, standard error of the mean.

evident in the absence of Dox, when leaky expression from the Dox-inducible promoter leads to a low amount of PTCH1 expression (*Figure 3B*). Under these conditions of low-level expression, the $t_{1/2}$ for cholesterol extraction by MβCD is ~27 s in cells expressing WT PTCH1 (*Figure 3C*) and ~57 s in cells expressing PTCH1-ΔL2 (*Figure 3I*).

If accessible cholesterol is the relevant substrate for PTCH1, expanding the pool of accessible cholesterol should increase the transport burden on PTCH1. Using signaling and differentiation assays, our prior work demonstrated that expanding the accessible cholesterol pool by SM depletion opposes PTCH1 activity: the dose of SHH required to activate signaling is reduced when accessible cholesterol levels are increased (*Kinnebrew et al., 2019*). In some cell lines, SM depletion even prevented PTCH1 from completely inhibiting SMO in the absence of SHH. To directly test this model, we measured the effect of SM depletion on the ability of PTCH1 to reduce MβCD-induced cholesterol extraction (*Figure 4A*). As demonstrated previously (*Figure 2F*), SM depletion increased both the extraction rate and total amount of cholesterol extracted by MβCD. When we expanded the pool of accessible cholesterol, PTCH1 was unable to completely prevent cholesterol extraction by MβCD. Importantly, PTCH1 was still active under these conditions since it reduced both the rate and total amount of cholesterol extracted (*Figure 4B*). These results show that SM depletion, which markedly potentiates Hh signaling (*Kinnebrew et al., 2019*), prevents PTCH1 from depleting the membrane outer leaflet of accessible cholesterol and provides an explanation for our previous observation of the opposing effects of PTCH1 and SM depletion on Hh signaling.

Taken together, these experiments show that PTCH1 depletes accessible cholesterol in the outer leaflet of the plasma membrane in a manner that is regulated by its endogenous ligand SHH.

## Requirement of transmembrane ion gradients for the function of PTCH1

PTCH1 has homology to the Resistance Nodulation Division (RND) family of pumps, which use TM proton gradients to efflux toxic molecules out of Gram-negative bacteria (*Tseng et al., 1999*). While a proton gradient does not exist across the plasma membrane of metazoan cells, PTCH1 is thought to

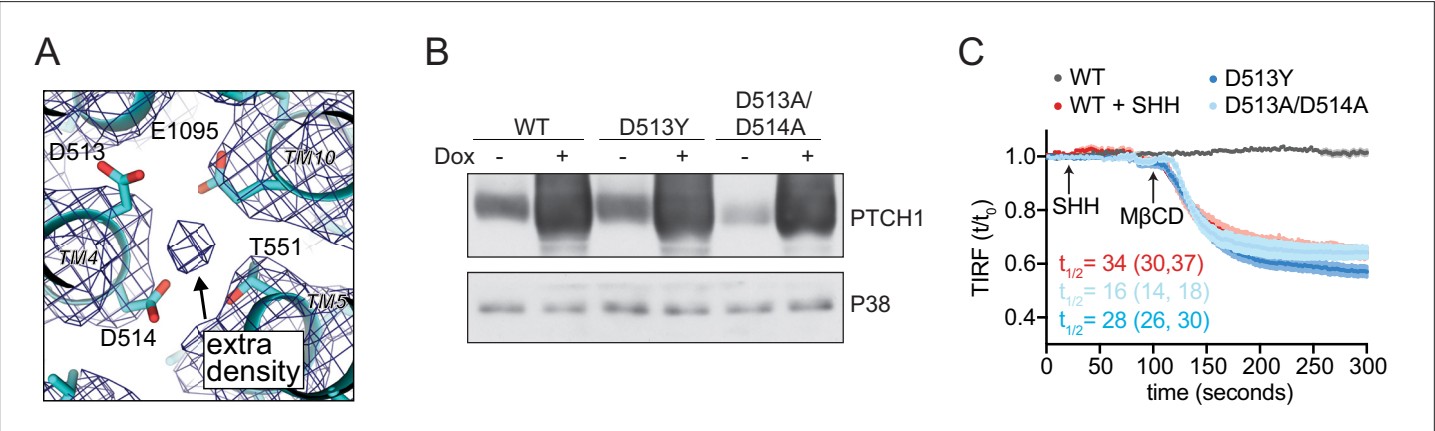

**Figure 5.** PTCH1 cation binding site mutants fail to reduce accessible cholesterol. (**A**) A view of the potential ion binding site in the middle of the PTCH1 transmembrane (TM) domain observed in the human SHH-PTCH1 complex (PDB 6DMY4) (*Gong et al., 2018*). The blue mesh represents the 3.6 Å cryo-EM map (EMD-7968) and shows an extra density, consistent with a potential ion, within binding distance to a cluster of negatively charged residues (D513, D514, and E1095). Two acidic TM residues (present within a GxxxDD motif in TM4 and GxxxE motif in TM10) are conserved in RND family proteins (*Petrov et al., 2020*; *Taipale et al., 2002*; *Zhang et al., 2018*). (**B**) Western blot showing Doxycycline (Dox)-inducible expression of PTCH1 wild-type (WT) or the D513Y and D513A/D514A mutants. (**C**) Time course of the change in TdTomato-PFOD4H membrane fluorescence after MβCD addition (316 μM, arrow) in cells expressing the indicated PTCH1 variants (+Dox). The inactivating effect of SHH (1 μM) on WT PTCH1 (red curve) is shown as a control. Each curve shows the mean fluorescence measured from >20 cells taken from at least three biological replicates, with SEM depicted in lighter shading around each curve. Fluorescence is depicted relative to the starting fluorescence ($t/t_0$) for the TIRFM data. The $t_{1/2}$, along with the 95% CI, is shown for each curve. The experiment was repeated three independent times with similar results. CI, confidence interval; SEM, standard error of the mean.

The online version of this article includes the following figure supplement(s) for figure 5:

**Source data 1.** Uncropped original scans of films used for immunoblots shown in *Figure 5B*.

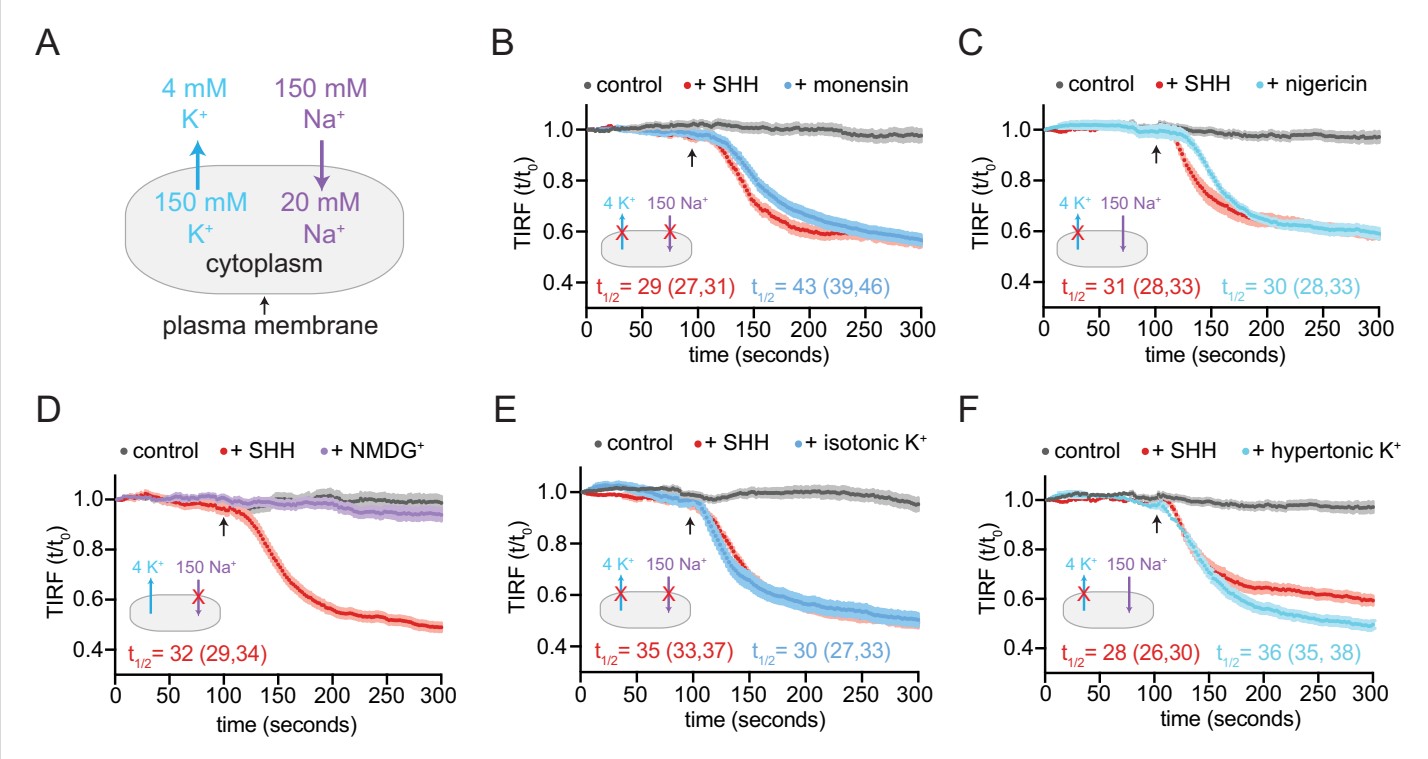

**Figure 6.** PTCH1 activity requires the cellular potassium gradient. (**A**) The sodium and potassium gradients present across the plasma membrane of mammalian cells. In each of the panels a red 'X' is used to mark the gradient that is disrupted by the treatment used in the panel. Panels (**B–F**) show time courses of the change in TdTomato-PFOD4H membrane fluorescence after MβCD addition (316 µM, arrow) in cells expressing PTCH1 in control media (gray curves) or in media with the indicated additions or substitutions. In each case, SHH has been added (1 µM, red curve) to show the effect of PTCH1 inactivation in control media. (**B, C**) Effect of monensin (**B**, 100 µM, 1 hr) or nigericin (**C**, 80 nM, 30 min). (**D, E**) Effect of isotonic media in which 150 mM NaCl is replaced with an equal concentration of NMDG-Cl (**D**) or KCl (**E**). (**F**) Effect of hypertonic media containing 150 mM KCl in addition to 150 mM NaCl. Each curve shows the mean fluorescence measured from >20 cells taken from at least three biological replicates, with SEM depicted in lighter shading around each curve. Fluorescence is depicted relative to the starting fluorescence ($t/t_0$) for the TIRFM data. The $t_{1/2}$, along with the 95% CI, is shown for each curve. The experiment was repeated three independent times with similar results. CI, confidence interval; SEM, standard error of the mean.

The online version of this article includes the following figure supplement(s) for figure 6:

**Figure supplement 1.** Effect of rubidium and cesium ions on PTCH1 activity.

depend on a cation gradient to power its cholesterol transport activity (***Ingham et al., 2000***; ***Taipale et al., 2002***). However, the identity of the cation gradient used by PTCH1 is disputed, with both the TM sodium (Na⁺) and potassium (K⁺) gradients being implicated (***Myers et al., 2017***; ***Petrov et al., 2020***). Neither of these studies used a direct assay for the biochemical activity of PTCH1; instead, they relied on SMO-induced changes in cAMP (***Myers et al., 2017***) or SMO accumulation in primary cilia (***Petrov et al., 2020***) as indirect readouts of PTCH1 activity.

PTCH1 has a triad of acidic residues (D513, D514, and E1095, residue numbers correspond to the human ortholog) that have been implicated in the ability of PTCH1 to inhibit SMO by coordinating the cation that permeates through the center of the TM domain to power its transport activity (***Petrov et al., 2020***; ***Taipale et al., 2002***; ***Zhang et al., 2018***). Structural studies have noted an extra cryo-EM density consistent with an ion directly coordinated by these residues (***Figure 5A***; ***Rudolf et al., 2019***). Mutations in these residues impair the ability of PTCH1 to block Hh signaling and have been linked to a familial cancer predisposition syndrome called Gorlin's syndrome (***Taipale et al., 2002***). Mutations of these key D513 and D514 residues abrogated the ability of PTCH1 to reduce outer leaflet accessible cholesterol, suggesting that the PTCH1 transport activity we observed in our TIRFM assay may also depend on cation-flux-driven conformational changes (***Figure 5B and C***). In addition, we were now in a position to address the uncertainty about the identity of the cation gradient required for PTCH1 function using a direct measure of its activity.

At rest in metazoan cells, the concentration of $Na^+$ is high on the outside and low on the inside; conversely, the concentration of $K^+$ is high inside and low outside (*Figure 6A*). The membrane potential (roughly –50 to –70 mV) is closest to the equilibrium potential of $K^+$ since metazoan membranes are most permeable to $K^+$. To test a role for cation gradients in PTCH1 sterol transport, we treated cells with the ionophoric antibiotic monensin, which non-selectively binds monovalent cations, transporting them across the cell membrane in an electroneutral (non-cell-depolarizing) manner. In the presence of monensin, PTCH1 failed to reduce accessible cholesterol in membranes, consistent with a role for monovalent cations in PTCH1 sterol transport (*Figure 6B*). PTCH1 activity was also blocked by treatment with nigericin, an ionophore that selectively transports $K^+$ ions across the cell membrane and thus dissipates the cellular $K^+$ gradient (*Figure 6C*).

To test the role of a $Na^+$ gradient, we incubated cells in a defined, isotonic extracellular medium containing either NaCl (control medium) or medium containing chloride salts of choline$^+$ or N-methyl-D-glucamine (NMDG$^+$). In comparison to $Na^+$, choline$^+$ and NMDG$^+$ are larger cations that are membrane impermeable at the shorter time scales used in our TIRFM assay. Replacement of extracellular $Na^+$ with choline$^+$ or NMDG$^+$ does not alter the $K^+$ gradient and has only a minor effect on the membrane potential (*Reuss and Grady, 1979*). PTCH1 functioned normally to reduce cholesterol accessibility when extracellular $Na^+$ was replaced with either NMDG$^+$ or choline$^+$, showing that the transport activity of PTCH1 does not depend on $Na^+$ permeation (*Figure 6D* and *Figure 6—figure supplement 1A*).

To test a role for the $K^+$ gradient, we used an isotonic extracellular medium in which NaCl was replaced with KCl, abolishing both the TM $K^+$ and $Na^+$ gradients. We also tested the effect of hypertonic medium containing physiological NaCl levels and high extracellular KCl, leaving the $Na^+$ gradient intact but abolishing the $K^+$ gradient. In both cases, the ability of PTCH1 to reduce accessible cholesterol was abolished in the presence of high extracellular $K^+$ (*Figure 6E and F*). Consistent with a previous study (*Petrov et al., 2020*), extracellular RbCl and CsCl also inhibited PTCH1 activity in our cholesterol transport assays (*Figure 6—figure supplement 1C*). Rb$^+$ and Cs$^+$, ions that are larger than $K^+$, may block the ion transport conduit through PTCH1 (*Petrov et al., 2020*). Taken together, these data show that PTCH1 requires the integrity of the TM $K^+$ gradient to reduce the accessibility of cholesterol in the outer leaflet of the plasma membrane.

## Discussion

The discovery that membrane cholesterol is segregated into different pools, made over two decades ago, was partly based on the observation that a fraction of cholesterol in both synthetic and cellular membranes is more susceptible to extraction by MβCD (*Haynes et al., 2000*; *Radhakrishnan and McConnell, 2000*). Subsequent characterization has shown that this fraction of membrane cholesterol is likely the same as the fraction that is more accessible to the cholesterol-modifying enzyme cholesterol oxidase and to cholesterol-binding toxins such as PFO and ALO (*Das et al., 2014*; *Gay et al., 2015*; *Lange and Steck, 2020*). These original studies established that the rate of cholesterol extraction by MβCD in both synthetic and cellular membranes serves as a sensitive and quantitative indicator of the size of the accessible cholesterol pool. Building on these previous studies, we established a live-cell, TIRFM assay to measure the impact of expressed proteins (like PTCH1) on accessible cholesterol in the outer leaflet of the plasma membrane. This assay allowed us to directly measure the biochemical activity of PTCH1 in the absence of downstream Hh signaling components like SMO. Our results show that PTCH1 depletes accessible cholesterol in the outer leaflet of the plasma membrane. Conversely, SHH inactivates PTCH1 and elevates outer leaflet cholesterol. More generally, our assay can be readily adapted to study the effect of any protein on membrane accessible cholesterol in the plasma membrane.

### The mechanism of cholesterol transport by PTCH1

The numerous PTCH1 cryo-EM structures solved to date give few insights into how PTCH1 may move cholesterol from the outer to the inner leaflet of the plasma membrane. Most of these structural studies have highlighted a hydrophobic tunnel through the extracellular domains of PTCH1 that extends to the outer leaflet of the plasma membrane (*Gong et al., 2018*; *Qi et al., 2018a*; *Zhang et al., 2020*). This tunnel has been proposed to provide a conduit for sterol transport based on the

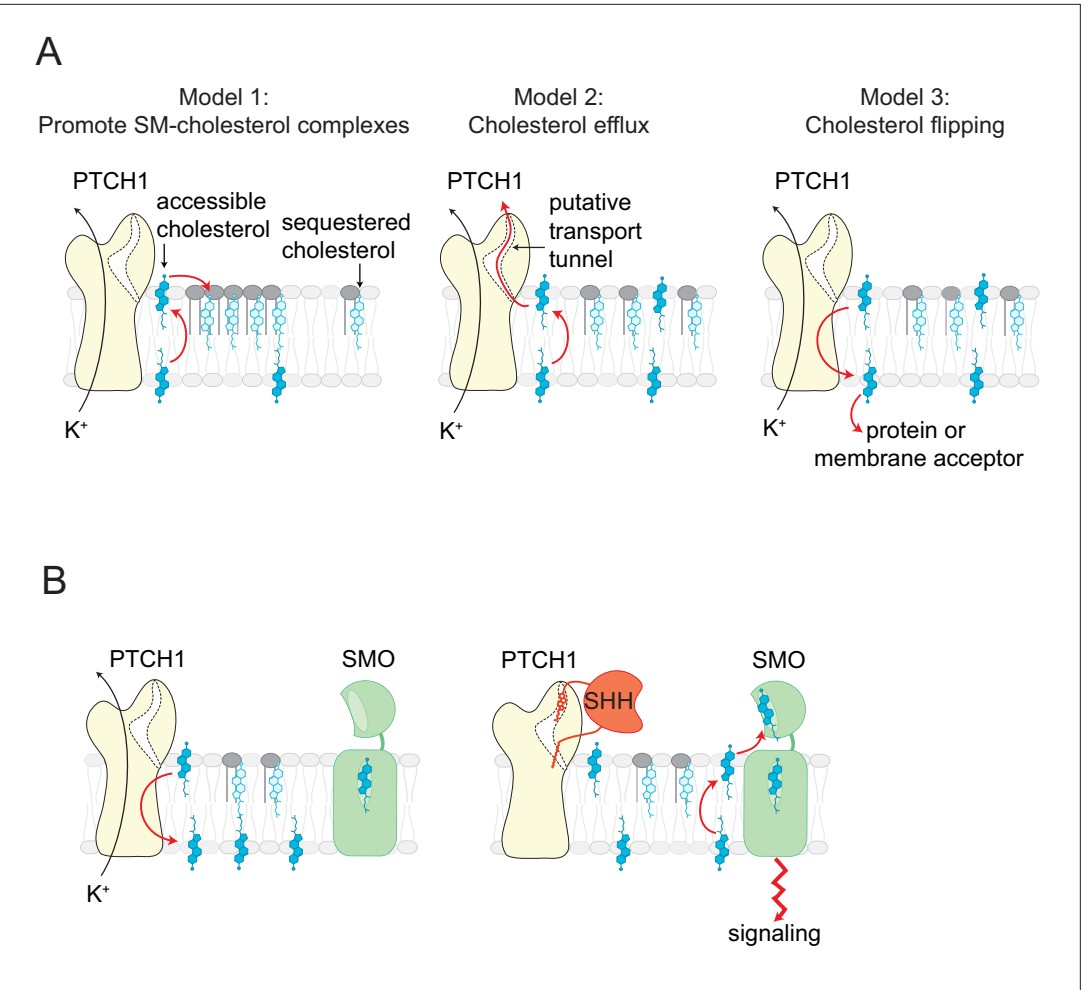

**Figure 7.** Models for how PTCH1 inhibits SMO by reducing membrane cholesterol. (**A**) Three proposed mechanisms by which PTCH1 depletes outer leaflet accessible cholesterol (see text for details). In models 1 and 2, inner leaflet cholesterol is predicted to be reduced, but in model three inner leaflet cholesterol will increase. (**B**) PTCH1 inhibits SMO by utilizing the cellular potassium gradient to transport outer leaflet accessible cholesterol to the cytoplasmic leaflet (left). When SHH binds to PTCH1, outer leaflet accessible cholesterol rises and activates SMO by binding to its extracellular cysteine-rich domain (right).

observation of sterol-like densities along its trajectory. Mutations predicted to block this tunnel inhibit PTCH1 activity. When SHH ligands bind to PTCH1, they insert both their attached lipids (the palmitate and the cholesterol) into this tunnel, presumably blocking sterol transport (**Qian et al., 2019**; **Qi et al., 2018b**; **Rudolf et al., 2019**). However, this transport route is inconsistent with two observations: (1) PTCH1-ΔL2 can suppress SMO activity and Hh signaling in cells and animals (**Briscoe et al., 2001**) and (2) PTCH1-ΔL2 can reduce outer leaflet cholesterol accessibility even more effectively than WT PTCH1 (**Figure 3—figure supplement 1F**). The excision of the entire second extracellular domain PTCH1, which forms one-half of the tunnel, should disrupt transport if the current models are correct. We speculate that the primary function of this PTCH1 tunnel may instead be to accommodate the palmitate and cholesterol moieties attached to SHH, thereby stabilizing an inactive state of the protein, while the TM domain plays a more direct role in inter-leaflet cholesterol flipping. Further structural, functional, and reconstitution studies are required to fully unravel the biochemical mechanism of this important developmental receptor and sterol transporter.

## The direction of cholesterol transport by PTCH1

Our observation that PTCH1 reduces accessible cholesterol in the outer leaflet of the plasma membrane raises an important question: what happens to this accessible cholesterol? Three possibilities are as

follows (*Figure 7A*): (1) PTCH1 alters the lateral organization of the membrane and promotes the formation of cholesterol-SM complexes in the outer leaflet, (2) PTCH1 transports this cholesterol out of the cell to an extracellular acceptor, or (3) PTCH1 moves the cholesterol to the inner leaflet or to an intracellular acceptor. Models 1 and 2 predict that PTCH1 should reduce inner leaflet cholesterol (due to the free exchange of cholesterol between the two leaflets), while model three predicts that PTCH1 would increase inner leaflet cholesterol (*Figure 7A*). We favor model three based on our observation that PTCH1 expression increases the recruitment of two different probes for accessible cholesterol to the membrane inner leaflet (*Figure 1*). We propose that PTCH1 uses potassium efflux to drive the movement of cholesterol from the outer to the inner leaflet of the plasma membrane, where it is likely then transported to other membrane-bound organelles, including the ER, by sterol transfer proteins. This directionality of transport by PTCH1 is the same as that used by the structurally related lysosomal cholesterol transporter NPC1.

## Comparison to alternative models for PTCH1 transporter function

Our results diverge from a previous study that proposed that PTCH1 depletes the inner leaflet of cholesterol, thereby preventing cholesterol from entering a tunnel through the center of the SMO transmembrane domain (TMD) (*Zhang et al., 2018*). Our TIRFM assay is conceptually different from the methods used in this previous study, where steady-state levels of inner leaflet cholesterol were measured by microinjecting cells with a mutant version of the PFOD4 domain that had been modified with environmentally sensitive fluorophores to lower the cholesterol threshold for binding (*Liu et al., 2017*). These modified PFOD4 probes do not differentiate between accessible and inaccessible pools of cholesterol since their binding to cholesterol-containing vesicles is insensitive to the presence of SM or other phospholipids (*Liu et al., 2017*). To avoid the complications associated with using these probes to measure membrane cholesterol accessibility (*Buwaneka et al., 2021*; *Courtney et al., 2018*; *Steck and Lange, 2018*), we sought to measure outer leaflet accessible cholesterol using a completely independent method. Our TIRFM assay uses the TdTomato-PFOD4H probe to simply monitor the cholesterol content of the inner leaflet; cholesterol accessibility in the outer leaflet is assessed by the rate of cholesterol transfer from the membrane to MβCD. The propensity of cholesterol to escape from membranes and transfer to a soluble acceptor like MβCD is directly related to its fugacity, which is a classical term for its chemical activity (*McConnell and Radhakrishnan, 2003*). The chemical activity of cholesterol can be markedly reduced by complex formation with membrane phospholipids like SM. Modulation of cholesterol's chemical activity by complex formation provides the basis for the accessible and inaccessible pools of cholesterol, which reflect free and complexed cholesterol, respectively. Our kinetic assay, grounded in previous experimental and theoretical work, is a more direct way to measure cholesterol accessibility compared to the steady-state binding of protein probes to membranes, which can be influenced by many extraneous factors (*Courtney et al., 2018*).

## Regulation of Smoothened by membrane cholesterol

Our findings have implications for some of the debates surrounding how SMO is activated in response to SHH. There is uncertainty about whether the cholesterol that binds and activates SMO comes from the inner or outer leaflet of the plasma membrane and about whether cholesterol activates SMO by binding to its extracellular cysteine-rich domain (CRD) or its TMD (*Byrne et al., 2016*; *Deshpande et al., 2019*; *Huang et al., 2016*; *Luchetti et al., 2016*; *Radhakrishnan et al., 2020*). Our observation that outer leaflet cholesterol levels rise in response to SHH favors the model that the source of cholesterol that activates SMO is the outer leaflet of the membrane. We propose that cholesterol from the outer leaflet binds to the CRD when PTCH1 is inactivated, driving SMO activation and Hh signaling (*Figure 7B*). An alternative model for the function of PTCH1 is that it inactivates SMO by directly accepting the cholesterol molecule bound to the SMO CRD (*Kong et al., 2019*). While our experiments did not test this model directly, our assays in HEK293T cells were performed without SMO co-expression. Thus, PTCH1 is able to change membrane cholesterol organization in the absence of SMO.

The inner and outer leaflets of the plasma membrane are different in their lipid composition (*Doktorova et al., 2020*; *Lorent et al., 2020*; *Verkleij et al., 1973*). Two factors suppress cholesterol accessibility in the outer leaflet: the exclusive presence of SM and the preponderance of phospholipids

with saturated acyl chains, which increase interactions with cholesterol (*Keller et al., 2000*; *Lönnfors et al., 2011*; *Slotte, 1992*). In contrast, the inner leaflet has higher levels of acyl chain unsaturation and lower packing, which would be predicted to increase cholesterol accessibility (*Lorent et al., 2020*). Indeed, our TdTomato-PFOD4H probe was readily recruited to the inner leaflet of the plasma membrane in the absence of any perturbations (*Figure 2C*). A recent independent study also confirmed that PFOD4-based probes bind to both leaflets of the plasma membrane of HeLa cells (*Abe and Kobayashi, 2021*). Given these considerations, we propose that the outer leaflet pool of cholesterol, especially in the SM-rich ciliary membrane (*Kinnebrew et al., 2019*), is a more plausible target of regulation by PTCH1 to control SMO activity. We emphasize that our model for PTCH1 function does not depend on (and does not inform) the outcome of current debates around the distribution of cholesterol between the inner and outer leaflets of the plasma membrane (*Buwaneka et al., 2021*; *Courtney et al., 2018*; *Liu et al., 2017*; *Steck and Lange, 2018*).

## Limitations of this study

A limitation of our assay is that it cannot monitor cholesterol extraction by MβCD selectively in the membrane of primary cilia—the compartment where PTCH1 is thought to inhibit SMO activity (*Rohatgi et al., 2007*). Consequently, we could not measure the effect of endogenously expressed PTCH1 on membrane cholesterol accessibility, since PTCH1 is largely concentrated in cilia. Instead, we used PTCH1 stably overexpressed throughout the plasma membrane of HEK293T cells, which likely has a different lipid and protein composition compared to the ciliary membrane. However, our previous studies in fixed cells demonstrated that PTCH1 inactivation leads to an increase in labeling of the outer leaflet of primary cilia by fluorescently labeled PFO* (*Kinnebrew et al., 2019*). Thus, both sets of studies support the model that inactivation of PTCH1 by SHH elevates accessible cholesterol in the outer leaflet of the plasma membrane.

An important question for future research is to understand the cation-driven conformational changes that allow cholesterol flipping by PTCH1. While the TM potassium gradient is required for PTCH1 function in cholesterol transport (based on our results) and SMO inhibition (*Petrov et al., 2020*), these studies did not demonstrate either the direct permeation of potassium through PTCH1 or the requirement of a potassium gradient in a purified assay of PTCH1 activity. Both studies were performed in the complex membrane environment of intact cells, which makes indirect effects impossible to exclude. This is a key issue because the potassium gradient across metazoan membranes is predicted to store less energy than the sodium gradient (since the resting membrane potential is closest to the equilibrium potential of potassium).

# Materials and methods

**Key resources table**

| Reagent type (species) or resource | Designation | Source or reference | Identifiers | Additional information |
|---|---|---|---|---|
| Genetic reagent (*Escherichia coli*) | BL21 Rosetta DE3 pLYS | MilliporeSigma | Cat# 71403-3 | |
| Cell line (*Homo sapiens*) | 293T-REx Flp-In | Invitrogen | Cat# R780-07; RRID:CVCL_U427 | |
| Cell line (*H. sapiens*) | 293T-REx Flp-In: PTCH1-1D4 | This paper | | Dr. Rajat Rohatgi (Stanford University) |
| Cell line (*H. sapiens*) | 293T-REx Flp-In: PTCH1-ΔL2-1D4 | This paper | | Dr. Rajat Rohatgi (Stanford University) |
| Cell line (*H. sapiens*) | 293T-REx Flp-In: PTCH1-1D4, D513Y | This paper | | Dr. Rajat Rohatgi (Stanford University) |
| Cell line (*H. sapiens*) | 293T-REx Flp-In: PTCH1-1D4, D513A/514 A | This paper | | Dr. Rajat Rohatgi (Stanford University) |

*Continued on next page*

*Continued*

| Reagent type (species) or resource | Designation | Source or reference | Identifiers | Additional information |
|---|---|---|---|---|
| Antibody | Anti-PTCH1 (Rabbit polyclonal) | PMID:17641202 | | Dr. Rajat Rohatgi (Stanford University) WB (1:500) |
| Antibody | Anti-RFP (Mouse monoclonal) | Thermo Fisher Scientific | Cat# MA5-15257; RRID:AB_10999796 | WB (1:2000) |
| Antibody | Anti-P38 (Rabbit polyclonal) | Abcam | Cat# ab7952; RRID: AB_306166 | WB (1:2000) |
| Antibody | Anti-mouse IgG (H + L) (Peroxidase AffiniPure Donkey) | Jackson ImmunoResearch Laboratories | Cat# 715-035-150; RRID: AB_2340770 | WB (1:10,000) |
| Antibody | Anti-rabbit IgG (H + L) (Peroxidase AffiniPure Donkey) | Jackson ImmunoResearch Laboratories | Cat# 111-035-144; RRID: AB_2307391 | WB (1:10,000) |
| Recombinant DNA reagent | mNeon-ALOD4 in pRSETB (plasmid) | PMID:33712199 | | Dr. Arun Radhakrishnan (University of Texas, Southwestern Medical Center) |
| Recombinant DNA reagent | mNeon-ALOD4- mutant in pRSETB (plasmid) | PMID:33712199 | | Dr. Arun Radhakrishnan (University of Texas, Southwestern Medical Center) |
| Recombinant DNA reagent | pcDNA5-FRT-TO Flp-In (plasmid) | Thermo Fisher Scientific | Cat# V652020 | |
| Recombinant DNA reagent | PTCH1-1D4 in pcDNA5-FRT-TO Flp-In (plasmid) | This paper | | Dr. Rajat Rohatgi (Stanford University) |
| Recombinant DNA reagent | dL2-PTCH1-1D4 in pcDNA5-FRT-TO Flp-In (plasmid) | This paper | | Dr. Rajat Rohatgi (Stanford University) |
| Recombinant DNA reagent | PTCH1-1D4, D513Y in pcDNA5-FRT-TO Flp-In (plasmid) | This paper | | Dr. Rajat Rohatgi (Stanford University) |
| Recombinant DNA reagent | PTCH1-1D4, D513A/D514A in pcDNA5-FRT-TO Flp-In (plasmid) | This paper | | Dr. Rajat Rohatgi (Stanford University) |
| Recombinant DNA reagent | TdTomato-PFO-D4H in pGEX-6P1 (plasmid) | PMID:25663704 | | Dr. Gregory Fairn (University of Toronto) |
| Recombinant DNA reagent | GFP-Gram$_{1b}$ in pEGFP-C1 (plasmid) | PMID:31724953 | | Dr. Yasunori Saheki (Nanyang Technological University, Singapore) |
| Recombinant DNA reagent | POG44 Flp-Recombinase expression vector (plasmid) | Thermo Fisher Scientific | Cat# V600520 | |
| Peptide, recombinant protein | Sonic Hedgehog | PMID:19561611 | | Dr. Christian Siebold (Oxford University) |
| Chemical compound, drug | Methyl beta cyclodextrin | Sigma-Aldrich | Cat# C4555-5G | |
| Chemical compound, drug | Myriocin | Cayman Chemicals | Cat# 35891-70-4 | |
| Chemical compound, drug | Doxycycline | Sigma-Aldrich | Cat# D9891 | |
| Chemical compound, drug | Monensin | Thermo Fisher Scientific | Cat# 461450010 | |

*Continued on next page*

*Continued*

| Reagent type (species) or resource | Designation | Source or reference | Identifiers | Additional information |
|---|---|---|---|---|
| Chemical compound, drug | Nigericin | Sigma-Aldrich | Cat# 481990 | |
| Chemical compound, drug | Dulbecco's modified Eagle's medium | Thermo Fisher Scientific | Cat# SH30081FS | |
| Chemical compound, drug | Fetal bovine serum | Sigma-Aldrich | Cat# S11150 | |
| Chemical compound, drug | Sodium pyruvate | Gibco | Cat# 11-360-070 | |
| Chemical compound, drug | L-glutamine | Gemini Bio-products | Cat# 400106 | |
| Chemical compound, drug | Penicillin/ streptomycin | Gemini Bio-products | Cat# 400109 | |
| Chemical compound, drug | Nonessential amino acids | Gibco | Cat# 11140076 | |
| Chemical compound, drug | Essential amino acids | Thermo Fisher Scientific | Cat# 11130051 | |
| Chemical compound, drug | Vitamins | Thermo Fisher Scientific | Cat# 11120052 | |
| Chemical compound, drug | Glucose | Thermo Fisher Scientific | Cat# A2494001 | |
| Chemical compound, drug | HEPES | Lonza | Cat# 17-737E | |
| Chemical compound, drug | Sodium pyruvate | Gibco | Cat# 11-360-070 | |
| Chemical compound, drug | HT supplement | Thermo Fisher Scientific | Cat# 11067030 | |
| Chemical compound, drug | Optimem | Thermo Fisher Scientific | Cat# 31985-070 | |
| Chemical compound, drug | Polyethylenimine | Polysciences | Cat# 23966-1 | |
| Chemical compound, drug | Matrigel | Thermo Fisher Scientific | Cat# CB-40234A | |
| Chemical compound, drug | SigmaFast Protease inhibitor cocktail, EDTA-free | Sigma-Aldrich | Cat# S8830 | |

*Continued on next page*

| Reagent type (species) or resource | Designation | Source or reference | Identifiers | Additional information |
|---|---|---|---|---|
| Chemical compound, drug | Hygromycin B | VWR Life Science | Cat# 97064-454 | |
| Chemical compound, drug | Nikon Immersion Oil, Type NF 50 cc, nd = 1.515 (23°C) | Nikon | Cat# MXA22024 | |
| Chemical compound, drug | Cesium chloride | Sigma-Aldrich | Cat# C4036 | |
| Chemical compound, drug | Rubidium chloride | Sigma-Aldrich | Cat# R2252 | |
| Chemical compound, drug | NMDG chloride | Sigma-Aldrich | Cat# 66,930 | |
| Chemical compound, drug | Choline chloride | Research Products International | Cat# C41040 | |
| Others | Chambered #1.5 German Coverglass, 8 well | Lab-Tek II | Cat# 155409 | |

## Constructs and plasmids

### PTCH1 constructs

Full-length WT mouse PTCH1, PTCH1-ΔL2, and the ion-binding site mutants (D513Y and D513A/D514A) were fused to a C-terminal 1D4 tag (amino acid sequence: TETSQVAPA) and cloned into the pcDNA5-FRT-TO Flp-In vector (Thermo Fisher Scientific, Cat# V652020) to enable inducible expression in the 293T-REx Flp-In cell system. PTCH1-ΔL2, which carries a deletion of the second extracellular loop (L2, amino acids 793–994), was a gift from James Briscoe (*Briscoe et al., 2001*).

### ALOD4 constructs

Plasmids encoding His6-mNeon-FLAG-ALOD4 (designated as ALOD4, mNeon-ALOD4, or WT ALOD4) and His6-mNeon-FLAG-ALOD4-mutant (designated as ALOD4-mutant) have been previously described (*Johnson and Radhakrishnan, 2021*). The ALOD4 used in these constructs (amino acids 404-512 of Anthrolysin O) contains two point mutations (S404C and C472A) that do not affect its activity. The cholesterol-binding mutant of ALOD4 contains six additional mutations (G501A, T502A, T503A, L504A, Y505A, and P506A) that abrogate cholesterol binding (*Endapally et al., 2019*). All constructs were cloned into the pRSETB vector for bacterial expression.

## Purification of SHH ligands

Human SHH carrying two isoleucine residues at the N-terminus and a hexahistidine tag at the C-terminus (known as SHH-C24II) was expressed in *Escherichia coli* Rosetta(DE3)pLysS cells and purified by immobilized metal affinity chromatography followed by gel filtration chromatography as described previously (*Bishop et al., 2009*). The two isoleucines at the N-terminus of SHH-C24II mimic the palmitate attached to native SHH. SHH-C24II, which otherwise lacks both the palmitate and cholesterol modification found in the native ligand, has been validated as an easy to purify and well-behaved substitute for the endogenous ligand (*Taylor et al., 2001*).

## mNeon-ALOD4 protein expression and purification

### Expression

ALOD4 plasmids were transformed into *E. coli* competent cells (BL21 [DE3] pLysS, MilliporeSigma, Cat #71403) and plated on Luria Broth (LB) agarose plates containing ampicillin (100 µg/ml) and chloramphenicol (34 µg/ml). A single colony was picked and grown overnight in 160 ml LB containing

ampicillin (100 µg/ml) and chloramphenicol (34 µg/ml) and used to inoculate a 1 L culture of LB. When the culture reached an $OD_{600}$ of 0.4–0.6, it was cooled to 18°C and protein expression was induced with 1 mM IPTG at 18°C for 18–24 hr. Cells were harvested at 3500×$g$ for 10 min at 4°C, and immediately used for purification.

## Purification

All steps of the purification were performed at 4°C or on ice. Cell pellets were resuspended in 20 ml of ice-cold Buffer A (50 mM Tris-HCl [pH 7.5], 150 mM NaCl, and 1 mM TCEP) supplemented with 1 mM phenylmethylsulfonyl fluoride (PMSF), 1× protease inhibitor (SigmaFast Protease inhibitor cocktail, EDTA-free; Sigma-Aldrich, Cat#S8830), and 1 mg/ml lysozyme and lysed by 10–15 passes through an EmulsiFlex C5 (Avestin) homogenizer. The Lysate was then clarified by centrifugation (30 min, 25,000×$g$ ) and incubated on an orbital shaker with 1 ml of Ni-NTA resin for 30 min. The protein-resin mixture was poured into a chromatography column to collect the flow through. The 1 ml packed column was washed with 50 ml of Buffer A and then 50 ml of Buffer A supplemented with 25 mM imidazole. Bound protein was eluted with Buffer A containing 250 mM imidazole in 5×1 ml fractions. Peak eluate fractions were pooled, concentrated (Amicon Ultra-4 10 kDa cutoff centrifugal filter), and loaded onto a Superdex 200 gel filtration column (Amersham Biosciences) equilibrated with Buffer A. Fractions containing pure His6-mNeon-Flag-ALOD4 (as judged by coomassie staining) were pooled, concentrated, aliquoted, and then stored in 20% glycerol at – 80°C. Immediately before use in experiments, an aliquot was thawed on ice, diluted to 5 µM in 0.5% serum Dulbecco's modified Eagle's medium (DMEM), and then added to live cells. See sections on Cell Culture and Drug Treatments for further details.

## Mammalian cell lines

293T-REx Flp-In cells were purchased from Invitrogen (Cat# R780-07). These purchased cell lines came with a certificate of authentication from the vendor and were used without further validation. All stable cell lines (see below) derived from the 293T-REx Flp-In cells were validated by Western blotting for the stably expressed protein. Cell lines were confirmed to be negative for *Mycoplasma* infection.

Cells were grown in high glucose DMEM (Thermo Fisher Scientific, Cat# SH30081FS) containing 10% fetal bovine serum (FBS) (Sigma-Aldrich, Cat# S11150) and the following supplements (hereafter called supplemented DMEM): 1 mM sodium pyruvate (Gibco, Cat# 11-360-070), 2 mM L-glutamine (Gemini Bio-products, Cat# 400106), 1× MEM nonessential amino acid solution (Gibco, Cat# 11140076), penicillin (40 U/ml), and streptomycin (40 µg/ml) (Gemini Bio-products, Cat# 400109). Supplemented DMEM was sterilized through a 0.2 µm filter and stored at 4°C. For TIRFM and Western blotting experiments, cells were seeded in 10% FBS supplemented DMEM and then grown to 90–100% confluency. Cells were serum starved in 0.5% FBS supplemented DMEM for 16 hr prior to the start of experiments.

## Stable cell line generation

293T-REx Flp-In cells (Invitrogen, Cat# R780-07) expressing PTCH1 were generated using PTCH1 constructs cloned into the pcDNA5 FRT-TO Flp-In vector. 293T-REx Flp-In cells were seeded in 10 cm cell culture dishes in 10% FBS supplemented DMEM lacking penicillin and streptomycin, grown to 75% confluency and transfected with a mixture of 750 ng of the pcDNA5 FRT-TO construct and 5.4 µg of the POG44 Flp-Recombinase expression vector (Thermo Fisher Scientific, Cat# V600520) using polyethyleneimine (PEI, linear transfection grade; Polysciences Cat# 23966-1). A ratio of 4 µl PEI: 1 µg of total DNA was prepared in 400 µl of room temperature OptiMEM (Thermo Fisher Scientific, Cat# 31985-070), mixed briefly by vortexing, and then incubated at room temperature for 15 min before adding dropwise to cells. After 24 hr, media was aspirated and replaced with 10% FBS supplemented media containing penicillin (40 U/ml) and streptomycin (40 µg/ml). After 24 hr, media was exchanged to fresh 10% FBS supplemented DMEM containing 2 µg/ml Hygromycin B (VWR Life Science, Cat# 97064-454). The majority of cells (>95%) died with Hygromycin B selection, and media was exchanged periodically to remove dead cells.

To confirm expression of PTCH1 variants, each respective cell line was seeded in 2× 6 cm dishes in 10% FBS supplemented DMEM. Cells were grown to 90–100% confluency and then media was exchanged to 0.5% FBS supplemented DMEM. In one 6 cm dish, PTCH1 expression was induced

by adding 1 µM Doxycycline. After 16–24 hr, cells were scraped off the plate in 4°C 1× phosphate-buffered saline and collected by centrifugation at 1000×*g*. Cells were lysed in 150 mM NaCl, 50 mM Tris-HCl pH 8, 10% NP-40, 1× protease inhibitor (SigmaFast Protease inhibitor cocktail, EDTA-free; Sigma-Aldrich, Cat#S8830), 1 mM $MgCl_2$, and 10% glycerol. Lysate was clarified by spinning at 20,000×*g* for 30 min and 4°C and then protein concentration was measured with a BCA assay. Equal amounts of protein were taken for each sample and incubated with 1 mM TCEP and 1× Laemmli buffer for 30 min at 37°C. Samples were then subjected to SDS/PAGE and finally blotted for expression of PTCH1 using a PTCH1 antibody raised against the PTCH1 cytoplasmic tail (*Rohatgi et al., 2007*) and P38 (anti-P38 rabbit polyclonal; Abcam, Cat# ab7952; RRID: AB_306166) as a loading control.

## Live cell imaging with total internal reflection fluorescence microscopy

### Cell preparation
All TIRFM experiments were carried out in live-cell imaging chambers (Lab-Tek II Chambered #1.5 German Coverglass, eight-well, Cat# 155409) on a Nikon TIRF system. To aid in cell adherence, the live cell imaging chambers were first prepared by coating with Matrigel (Corning 356234). Matrigel was diluted 1:20 in ice-cold 0.5% FBS supplemented DMEM and 100 µl was added to coat the bottom of each well. After allowing Matrigel to solidify for 40 min at room temperature, chambers were washed once with 37°C 10% FBS supplemented DMEM prior to cell plating.

Cells were counted and plated in 300 µl per well at a density of 90,000 cells per well. After 24 hr, or at 90% confluency, cells were transfected with TdTomato-PFOD4H. A transfection reaction for one well included 100 ng TdTomato-PFOD4H and 0.4 µl PEI diluted (ratio of 4 µl PEI: 1 µg of total DNA) into 15 µl of OptiMEM (Thermo Fisher Scientific, Cat# 31985-070) (typically at least eight wells were transfected simultaneously, and transfection reactions were prepared in one 8× mixture). Transfection reactions were briefly vortexed and then incubated at room temperature for 15 min. The transfection reaction was then diluted into 37°C 0.5% FBS supplemented DMEM at a sufficient volume to add 300 µl to each well of the live-cell chamber. Finally, the media from each well-containing cells was aspirated, and 300 µl of transfection mixture was added. If PTCH1 expression was desired, doxycycline was added to a final concentration of 1 µM directly to the diluted transfection mixture. Approximately 16 hr later, live-cell imaging was performed. Immediately before imaging, culture media was replaced with 300 µl 37°C 0.5% FBS supplemented DMEM.

### Microscopy
The TIRF microscope live-cell imaging chamber was warmed to 37°C prior to imaging. Cells were imaged with a Nikon Apo TIRF 100×/1.49 Oil objective (W.D. 0.12, coverglass thickness 0.13–0.2). TdTomato-PFOD4H fluorescence was excited with a 561 nm laser and GFP-GRAM$_{1b}$ fluorescence was excited with a 491 nm laser. All imaging was carried out using MicroManager software (https://micro-manager.org/). First, oil was added to the objective (Nikon Immersion Oil, Type NF 50 cc, nd = 1.515 (23°C); Cat# MXA22024) and then cells were located. An optimum TIRF angle was established. Multidimensional acquisition parameters were set such that an image was collected every 2 s, for 150 frames (300 s total). Once a movie was initiated, SHH was added at 1 µM at 20 s (frame 10) when indicated, and MβCD was added at 316 µM (unless otherwise stated) at 100 s (frame 50).

Steady-state measurements of TdTomato-PFOD4H and GFP-GRAM$_{1b}$ were captured by first locating cells and identifying an optimum TIRF angle. Fields of cells were selected if they had at least five cells. Fluorescence was captured using the MicroManager software 'Snap Shot' function, and then each image was saved for downstream analysis (see next section).

### TIRFM analysis
TIRFM movies were analyzed in Fiji2 using a Time Series Analyzer plugin (https://imagej.nih.gov/ij/plugins/time-series.html). To quantify changes in TdTomato-PFOD4H fluorescence, a region of cell membrane was selected that made up ~20–50% of the total surface area of a cell. Each field of cells contained more than five cells, enabling >5 measurements per movie. At least three movies were generated per treatment condition and each experiment was repeated at least three independent times on separate days. Once an area to be measured was chosen, it was added to the Time Series Analyzer window. The average fluorescence of each selected area was then calculated using the 'Get

Average' function, returning a matrix containing the average fluorescence values for each selected region at each time point for a given movie. The average fluorescence values for each chosen area were then normalized to the starting average fluorescence ($t/t_0$), such that the starting fluorescence value equaled one. Finally, values were plotted in GraphPad Prism 9.1 with time in seconds on the x-axis and $t/t_0$ average fluorescence values on the y-axis. Raw curves, without normalization to the baseline, are shown for selected experiments in *Figure 2—figure supplement 1D* and *Figure 3—figure supplement 2*. Each circle making up a point on a curve represents the mean TdTomato-PFOD4H fluorescence at that time point, and the standard error of the mean is depicted as the shading around the curve. The number of cells measured for each experiment is stated throughout the text in the figure legends. Curve fitting was performed in GraphPad Prism 9.1 using the nonlinear regression curve fit for a 'plateau followed by a one phase exponential decay.' The time taken to reach half-maximal fluorescence ($t_{1/2}$) for each curve is reported in the figures, with the upper and lower 95% confidence interval bounds denoted in parentheses.

Analysis of steady-state TdTomato-PFOD4H and GFP-GRAM$_{1b}$ fluorescence was carried out in Fiji2. A region of cell membrane was selected that made up ~20–50% of the total surface area of the cell. The average signal intensity for that region was measured with the 'Ctrl-m' function of Fiji2. Values were then plotted without normalization in GraphPad Prism 9.1 using a column graph showing individual values. Outliers were excluded using the Identify Outlier function of GraphPad Prism 9.1 (ROUT method with a Q-score=10%).

## Drug treatments for total internal reflection fluorescence microscopy

### MβCD treatment

Methyl-β-cyclodextrin (MβCD) (Sigma-Aldrich, Cat# C4555-5G) was diluted immediately prior to experiments in filtered milliQ water to generate a 38 mM stock. During imaging, MβCD is added directly to cells at indicated concentration without prior dilution.

### ALOD4 treatment

To deplete accessible cholesterol with ALOD4, cells were seeded in 10% FBS supplemented DMEM and grown to 90–100% confluency. Cells were then transfected with TdTomato-PFOD4H in 0.5% supplemented DMEM (see section on Live-cell imaging with TIRFM) for 16 hr. Finally, low serum supplemented DMEM was exchanged for 0.5% supplemented DMEM containing 5 µM mNeon-ALOD4 or 5 µM of the mNeon-ALOD4 cholesterol-binding mutant. Cells were then returned to the cell culture incubator ( 37°C and 5% CO2) for 45 min. Immediately before imaging, media containing ALOD4 was removed and replaced with 37°C 0.5% supplemented DMEM.

### Sphingomyelin depletion with myriocin

Cells were seeded in 10% FBS supplemented DMEM at an initial density that allowed for 3 days of growth prior to experimentation. After the cells adhered to Matrigel-coated live cell imaging chambers (1–2 hr), media was removed and replaced with 10% FBS supplemented DMEM containing 80 µM myriocin. 16–24 hr before imaging, cells were transfected with TdTomato-PFOD4H (see section on Live-cell imaging with TIRFM) in 0.5% FBS supplemented DMEM containing fresh 80 µM myriocin.

### Nigericin and monensin treatments

Prior to live-cell imaging, media were removed from each well of the live-cell imaging chambers and replaced with 37°C 0.5% FBS supplemented DMEM containing 80 nM nigericin (Sigma-Aldrich, Cat# 481990) for 30 min or 100 µM monensin (Thermo Fisher Scientific, Cat# 461450010) for 1 hr.

### Ion gradient treatments

To test the effect of various ions, Base Media was prepared with the following components: 1× essential amino acids (Thermo Fisher Scientific, Cat# 11130051), 1× MEM nonessential amino acid solution (Gibco, Cat# 11140076), 2 mM L-glutamine (Gemini Bio-products, Cat# 400106), 1× vitamins (Thermo Fisher Scientific, Cat# 11120052), 1× glucose (Thermo Fisher Scientific, Cat# A2494001), 50 mM HEPES (Lonza, Cat# 17-737E), 1 mM sodium pyruvate (Gibco, Cat# 11-360-070), 1× HT supplement (Thermo Fisher Scientific, Cat# 11067030), 2 mM $CaCl_2$, 1 mM $MgSO_4$, and 0.5% FBS. Base Media was sterilized through a 0.2 µm filter and stored at 4°C. Immediately prior to experiments, media

was supplemented with salts at specified concentrations denoted in the figure legends. For isotonic control media, NaCl was added to a final concentration of 150 mM and KCl was added to 5 mM.

## Statistical analysis

Data analysis and visualization were performed in GraphPad Prism 9.1. Model figures (*Figures 1A, 2A,B, 3–4A, 6A, 7A and B*) were made in Adobe Illustrator CS6. TIRFM and epifluorescence images (*Figure 2C*) were rendered in Fiji2 and the Patched one structure (*Figure 5A*, PDB 6DMY4) was generated in PyMOL. Scatter dot plots (*Figure 1B–C*, *Figure 1—figure supplement 1A* and *Figure 2—figure supplement 1A, B*) were generated with default settings in GraphPad Prism 9.1; outliers were excluded using the Identify Outlier function of GraphPad Prism 9.1 (ROUT method with a Q-score=10%). Median and interquartile ranges for each plot are denoted by horizontal and vertical lines, respectively.

All statistical analyses used nonparametric methods, which do not assume an underlying normal distribution in the data. The statistical significance of differences between two groups (*Figure 1B–C*, *Figure 1—figure supplement 1A* and *Figure 2—figure supplement 1A, B*) was determined by the Mann-Whitney test. Information about error bars, statistical tests, p-values and n values are reported in each figure legend and were calculated using GraphPad Prism 9.1. All experiments included at least three independent trials with consistent results, unless otherwise noted in the figure legend.

Throughout the paper, the numerical p-values for the comparisons from GraphPad Prism 9.1 are given in the figure legends and denoted on the graphs according to the following key: $*p \leq 0.05$, $**p \leq 0.01$, $***p \leq 0.001$, $****p \leq 0.0001$, non-significant (ns) $p > 0.05$.

## Acknowledgements

The authors thank Chandni Patel and Hermann Broder Schmidt for help on experiments (Figure 6) to alter transmembrane ion gradients, Greg Fairn for providing the TdTomato-PFOD4H probe and Kristen Johnson, Ted Steck, and Yvonne Lange for helpful discussions.

## Additional information

### Competing interests

Arun Radhakrishnan: is a reviewing editor for eLife. The other authors declare that no competing interests exist.

### Funding

| Funder | Grant reference number | Author |
|---|---|---|
| Cancer Research UK | C20724 | Christian Siebold |
| Cancer Research UK | A26752 | Christian Siebold |
| European Research Council | 647278 | Christian Siebold |
| National Institutes of Health | GM118082 | Rajat Rohatgi |
| National Institutes of Health | GM106078 | Rajat Rohatgi |
| National Institutes of Health | HL20948 | Arun Radhakrishnan |
| Welch Foundation | I-1793 | Arun Radhakrishnan |
| Leducq Foundation | 19CVD04 | Arun Radhakrishnan |
| Ministry of Education, Singapore | MOE2017-T2-2-001 | Yasunori Saheki |

| Funder | Grant reference number | Author |
|---|---|---|
| Ministry of Education, Singapore | MOE-T2EP30120-0002 | Yasunori Saheki |
| National Science Foundation | Predoctoral Fellowship | Maia Kinnebrew |
| Ford Foundation | Predoctoral Fellowship | Giovanni Luchetti |

The funders had no role in study design, data collection and interpretation, or the decision to submit the work for publication.

## Author contributions

Maia Kinnebrew, Conceptualization, Data curation, Formal analysis, Investigation, Methodology, Validation, Writing - original draft, Writing – review and editing; Giovanni Luchetti, Investigation, Methodology, Writing – review and editing; Ria Sircar, Investigation, Methodology, Resources; Sara Frigui, Investigation, Writing – review and editing; Lucrezia Vittoria Viti, Investigation, Methodology; Tomoki Naito, Methodology, Resources; Francis Beckert, Methodology, Software; Yasunori Saheki, Methodology, Resources, Supervision, Writing – review and editing; Christian Siebold, Formal analysis, Methodology, Resources, Supervision, Writing – review and editing; Arun Radhakrishnan, Conceptualization, Formal analysis, Methodology, Resources, Supervision; Rajat Rohatgi, Conceptualization, Funding acquisition, Project administration, Resources, Supervision, Writing - original draft, Writing – review and editing

## Author ORCIDs

Maia Kinnebrew (iD) http://orcid.org/0000-0002-7344-8231
Tomoki Naito (iD) http://orcid.org/0000-0002-8393-3601
Yasunori Saheki (iD) http://orcid.org/0000-0002-1229-6668
Christian Siebold (iD) http://orcid.org/0000-0002-6635-3621
Arun Radhakrishnan (iD) http://orcid.org/0000-0002-7266-7336
Rajat Rohatgi (iD) http://orcid.org/0000-0001-7609-8858

## Decision letter and Author response

Decision letter https://doi.org/10.7554/eLife.70504.sa1
Author response https://doi.org/10.7554/eLife.70504.sa2

# Additional files

## Supplementary files

• Transparent reporting form

## Data availability

No dataset was generated or used during this study (such as deep sequencing data, mass spectrometry data, structural coordinates or maps, genetic data or clinical trial data) that required deposition in a repository such as GenBank, the PDB, mass spec data repositories, or clinical data repositories. We have provided original, uncropped scans of immunoblots shown in Figures 2B, 4B, and Figure 3-figure supplement 1 in the Source Data Files. All other data generated are included in this study, with replicates and statistics described in the figure legends and methods.

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
