## [Editor Report]

This paper addresses an important question regarding the mechanisms of Hedgehog signaling. The authors develop a new method to observe changes in cholesterol accessibility in the outer lamella of the plasma membrane to investigate the activity of the Hedgehog receptor PTCH1 and its modulation by Sonic Hedgehog. The results support the conclusion that PTCH1 needs a potassium gradient to reduce chemically-active cholesterol in the outer lamella, presumably by translocating the sterol to the inner lamella. The proposed model contradicts previous reports that suggest transport in the opposite direction using the plasma membrane sodium gradient for energy. In the initial review, the reviewers appreciated the potential impact of the findings and suggested several areas for improvement. The authors have now satisfactorily addressed the reviewers' comments in the revised manuscript.

---

## [Decision Letter]

**Decision letter after peer review:**

Thank you for sending your article entitled "Patched 1 reduces the accessibility of cholesterol in the outer leaflet of membranes" for peer review at *eLife*. Your article is being evaluated by 2 peer reviewers, and the evaluation is being overseen by a Reviewing Editor and Marianne Bronner as the Senior Editor.

This paper addresses an important question regarding the mechanisms of Hedgehog signaling. The authors develop a new method to observe changes in cholesterol accessibility in the outer lamella of the plasma membrane to investigate the activity of the Hedgehog receptor PTCH1 and its modulation by Sonic Hedgehog. The results support the conclusion that PTCH1 needs a potassium gradient to reduce chemically-active cholesterol in the outer lamella, presumably by translocating the sterol to the inner lamella. The proposed model contradicts previous reports that suggest transport in the opposite direction using the plasma membrane sodium gradient for energy. While the reviewers appreciate the potential impact of the findings, a major limitation of this study is the reliance on overexpressed PTCH1. The reviewers and editors feel that demonstrating a similar activity for endogenous PTCH1, for example by comparing endogenous PTCH1 with or without SHH ligand or Ptch1+/+ vs Ptch1-/- cells, is essential.

*Reviewer #1 (Recommendations for the authors):*

This manuscript investigates how the Hedgehog receptor protein Patched 1 (PTCH1) controls signaling activity of the transducer protein Smoothened (SMO). Studies follow up on a previous *eLife* publication from Rohatgi and Seibold labs suggesting PTCH1 represses SMO by limiting accessible cholesterol in cellular membranes. Cholesterol is proposed to function as the SMO agonist through binding a site in its extracellular cysteine rich domain (CRD), and that PTCH1 governs SMO activation by preventing accessible membrane cholesterol from entering this site.

A pro of the study is the development and validation of a quantitative fluorescent assay to monitor available membrane cholesterol. This assay measures accessible cholesterol through TIRF imaging of a fluorescent cholesterol binding protein (PFO) that associates with accessible (non-sphingomyelin associated) membrane cholesterol. In each experiment, MβCD is used to extract cholesterol from the outer membrane, which leads to flipping of cholesterol from inner to outer plasma membrane. Depletion of cholesterol from the inner membrane releases PFO to the cytoplasm, which decreases TIRF signal. This assay is the first to measure 'enzymatic' activity by PTCH1 in a quantifiable way. In prior studies, PTCH1 control of membrane cholesterol was inferred by tracking SMO ciliary localization, which is inhibited by active PTCH1. In addition to this SHH-centric benefit, the assay will likely be useful for other research applications necessitating quantification of available membrane cholesterol.

Using the PFO TIRF assay, the authors provide evidence that PTCH1 reduces accessible cholesterol and that the ligand SHH blocks this activity. Use of additional cholesterol binding probes allowed the authors to propose a model in which PTCH1 flips cholesterol from the outer to inner plasma membrane. They speculate that inner leaflet cholesterol is then transported by an intracellular cholesterol binding protein to shuttle it away from the membrane, thus preventing its association with SMO.

Despite the methodological advance of the PFO TIRF assay, the study only modestly advances knowledge beyond what has already been reported by this group and others – that PTCH1 moves cholesterol (Rohatgi, Seibold, Beachy, Salic Labs) and that it requires the K^+^ gradient to do so (Salic Lab). However, since the manuscript is being considered as an *eLife* Research Advance, rather than a Research Article, I am supportive of publication.

I don't disagree with the data interpretations but am having a hard time visualizing the biology of the system when PTCH1 is expressed because of the way the MβCD results are presented. The gist is that there is less free cholesterol in the outer membrane when PTCH1 is around, so MβCD doesn't have much to extract and PFO TIRF is not redistributing. However, since the lines don't shift following MβCD (Figure 2C as an example), it looks like cholesterol is locked in the membrane by PTCH1, rather than already having been depleted. I think this is a visualization problem because everything is normalized to 1. Are the baseline readings significantly different between the samples? Is baseline higher/lower when PTCH1 is expressed? I needed to sketch it out for myself, and I feel like this is too much work for the reader. Is there a way to convey the results so they're more intuitive? More cartoons, maybe? The cartoons provided are very helpful.

Re: Figure 6 and related to the point above: If PTCH1 is flipping cholesterol to the inner membrane, this should be reflected by PFO TIRF even in the absence of MβCD. Do you see a higher baseline in response to Dox induction of PTCH1 expression? The results presented in Figure 6 are less convincing that the rest of the figures, so it felt like the manuscript was ending with an afterthought. This point would be improved by additional experimental support and more discussion. There is only one sentence discussing the results in Figures 6B and C.

*Reviewer #2 (Recommendations for the authors):*

Recent publications suggest that PTCH1 is a cholesterol transporter that mobilises cholesterol or a cholesterol derivative from the inner leaflet of the plasma membrane to the outer leaflet, where its concentration is about 10-fold higher (Zhang et al., 2018, Cell 175, 1352-1364). This transport against a concentration gradient requires energy, and previous literature indicates that sodium is the cation that provides such energy (Myers et al., Proc Natl Acad Sci U S A 2017;114(52):E11141-E11150). PTCH1 contains an acidic triad (EED), conserved in bacterial RND permeases, that is essential for its activity as Smoothened repressor and likely allows co-transport of the cation.

In this study, Kinnebrew et al., follow their previous finding that indicates that sphingomyelin (SM) depletion potentiates Hedgehog signalling (Kinnebrew et al., *eLife*. 2019 Oct 30;8:e50051). SM localises in the outer leaflet of the plasma membrane and forms a complex with cholesterol, reducing availability of "free" or chemically-active cholesterol. Given that this suggests that increases in cholesterol concentration in the outer leaflet drives Smoothened activation, as opposed to the findings of the Beachy Lab, they re-evaluate PTCH1's transport directionality in the current study using total internal reflection fluorescence microscopy (TIRFM) to image reduction of binding of a cholesterol sensor domain fused to a fluorescent protein (TdTomato) to the inner leaflet when cholesterol is extracted from the outer lamella using methyl-β-cyclodextrine. The findings support the author's conclusion that PTCH1 reduces accessible outer leaflet cholesterol, but several methodological and conceptual questions remain:

– MβCD pinches accessible cholesterol out of the membrane, but it is unclear if the rate of extraction is comparable to the unknown rate of transport by PTCH1, or if a rate-limiting aspect of the model confounds the interpretation of the rate of PFOD4H-TdTomato fluorescence reduction. Therefore, while the assay can be optimal to capture spontaneous flip-flopping, it may not be a faithful readout of a facilitated transport event.

– PFOD4H-TdTomato fluorescence in the steady-state is more likely to represent the inner leaflet cholesterol content. Because the TIRFM signals are normalised to t=0, this crucial information is not available in most experiments other than in Figure 6. This would be particularly useful in cells expressing PTCH1 and PTCH1-DL2, and after a few minutes of SHH addition, as the affinity of PFOD4H for cholesterol is close to the reported inner leaflet cholesterol level (~ 1-3 mol%).

– If myriocin treatment increases accessible cholesterol in the outer leaflet, wouldn't it increase the inner leaflet cholesterol content by flip-flop? If that is the case, the non-normalised TIRF signal should be higher at t=0 than in vehicle-treated cells.

– The concentration of SHH ligand used to module cholesterol availability seems excessively high (10-6 M). The same group used 25 nM in the previous study as a "high, saturating SHH concentration", in line with most groups using SHH in the range of 50-100 nM. A dose-response effect will also inform if the IC50 of SHH in the TIRF assay agrees with its Kd for PTCH1 and the EC50 for stimulation of Gli-dependent transcription.

– The lipid modifications of SHH play a key role in the asymmetric binding mode to PTCH1 dimers. The article does not detail the modifications of the SHH used.

– Induction of PTCH1 expression increases absolute TIRF signals in Figure 6, suggesting higher inner leaflet cholesterol. However, this interpretation depends on equal total fluorescence, i.e. equal expression of the fluorescent sensor proteins regardless of PTCH1 expression and SHH signalling.

– If PTCH1 reduces accessible cholesterol in the outer lamella, one would expect to observe a reduction in binding of an extracellular cholesterol sensor, equivalent to the PFOD4H-TdTomato.

– The biggest limitation of this study is the reliance on overexpressed PTCH1. The effect of SHH addition to Hh-competent cells expressing endogenous PTCH1 (such as NIH 3T3 cells) on cholesterol sensor measurements is essential, even if it cannot be determined in the primary cilium membrane.

– Conceptual concern: why would PTCH1 require energy provided by a cation gradient to move cholesterol in favour of its concentration gradient? The related cholesterol transporter NPC1 transports cholesterol from the outer to inner leaflet without an energy gradient, and it lacks the acidic triad essential for PTCH1 activity.

– This study clashes with previous reports using confocal microscopy vs TIRF. One question that comes to mind is if a different conclusion would be drawn using the PFOD4H-TdTOmato sensor in standard confocal imaging to image changes in lateral membranes that are more readily exposed to solvent and SHH than the basal membrane attached to the substratum.

– The contradictory findings are unlikely to be explained simply by a difference in the cholesterol sensor used (PFOD4H-TdTOmato vs PFOD4H tagged with a small solvatochromic fluorophore). Additional controls and a deeper discussion of the geometry of the assay and the potential impact of measuring MβCD-induces changes in fluorescence vs steady-state levels will be necessary to understand the differences.

However, I identified some areas that need to be addressed. I would like you to consider providing additional evidence for some of my key concerns:

1 – Provide baseline absolute levels of fluorescence in each condition.

2 – Demonstrate that induction of PTCH1 does not affect total expression of PFOD4H-TdTomato.

3 – Investigate the effect of adding more physiological concentrations of Shh (in the 10-50 nM range) to Hh-competent cells expressing the cholesterol sensor. Potential controls of acute silencing of PTCH1.

4 – Testing changes in binding of an extracellular cholesterol sensor in the system.

5 – Indicate source and molecular details of the SHH ligand used: is it lipidated, is the Ile-Ile mimic of the N-terminal palmitate? Please add this to the methods.

If Shh is unmodified, could you test if dual lipidated SHH has the same effects at a lower concentration?

6 – I'd also like to see a deeper discussion on the potential explanations of the Zhang et al., results. You are creating a lively controversy in the field and would strongly benefit from trying to figure out why your conclusions are correct and the other is not when the systems are so similar in most ways.

---

## [Author Response]

This paper addresses an important question regarding the mechanisms of Hedgehog signaling. The authors develop a new method to observe changes in cholesterol accessibility in the outer lamella of the plasma membrane to investigate the activity of the Hedgehog receptor PTCH1 and its modulation by Sonic Hedgehog. The results support the conclusion that PTCH1 needs a potassium gradient to reduce chemically-active cholesterol in the outer lamella, presumably by translocating the sterol to the inner lamella. The proposed model contradicts previous reports that suggest transport in the opposite direction using the plasma membrane sodium gradient for energy. While the reviewers appreciate the potential impact of the findings, a major limitation of this study is the reliance on overexpressed PTCH1. The reviewers and editors feel that demonstrating a similar activity for endogenous PTCH1, for example by comparing endogenous PTCH1 with or without SHH ligand or Ptch1+/+ vs Ptch1-/- cells, is essential.

This important point, which is the basis of our current manuscript, has been demonstrated in Figure 8 of the parental paper for this *Research Advance* (Kinnebrew et al., 2019). A key fact relevant to this question is that the bulk of endogenous PTCH1 in NIH/3T3 cells is localized in the ciliary membrane, a miniscule subcompartment of the plasma membrane (Rohatgi et al., 2007). In our prior study, we used the steady-state binding of fluorescently-labeled PFO* to measure outer leaflet accessible cholesterol either over the whole plasma membrane (by FACS, Figure 8A) or selectively at primary cilia (by microscopy, Figure 8B) in NIH/3T3 cells expressing endogenous PTCH1. SHH induced an increase in outer leaflet PFO* binding at the ciliary membrane (Figure 8B), consistent with the data from TIRFM assays presented in the current manuscript. Importantly, SHH did not change PFO* binding to the overall plasma membrane (Figure 8A) in NIH/3T3 cells, since PTCH1 localizes and functions at primary cilia (not all over the plasma membrane). Our TIRFM assay reports on the entire plasma membrane, not specifically on the ciliary membrane. Consequently, our TIRFM assay does not detect any changes in outer leaflet cholesterol accessibility in NIH/3T3 cell or Mouse Embryonic Fibroblasts (MEFs) after treatment with SHH (a result completely consistent with our 2019 *elife* paper showing that PFO* binding to the outer leaflet of the plasma membrane in NIH/3T3 cells is unchanged by SHH). In conclusion, the results of steady-state PFO* binding in NIH/3T3 cells shown in our 2019 *elife* paper are in agreement with the TIRFM assay used in the current manuscript. In the context of both endogenous PTCH1 (2019 paper) and overexpressed PTCH1 (current manuscript), SHH causes an increase in outer leaflet cholesterol accessibility.

We agree that using endogenously expressed proteins is preferable whenever possible. However, we respectfully take the position that model assays using overexpression in heterologous systems have played an irreplaceable role in elucidating the molecular mechanism of many transporter proteins and channels (whose biochemical activities would have been otherwise very difficult to directly characterize due to low or highly compartmentalized expression). A prominent example is the extensive use of mRNA injection in *Xenopus laevis* oocytes to overexpress and then characterize functions of diverse mammalian and even plant membrane proteins (Pike et al., 2019; Wagner et al., 2000). Other examples include the use of overexpression in HEK293T and Cos cells to study the mechanism of other cholesterol/sterol transporters, including ABC1 and the scavenger receptor SR-BI (Acton et al., 1996; Wang et al., 2001). Perhaps the most relevant example in the context of our manuscript are studies that demonstrated that PTCH1 is the direct receptor for SHH. Biochemical evidence that SHH binds to PTCH1 required the use of overexpression: in *Xenopus laevis* oocytes by Cliff Tabin’s lab (Marigo et al., 1996) or HEK293 cells by Arnon Rosenthal’s lab (Stone et al., 1996). Due to low endogenous PTCH1 expression levels, it is impossible to detect differences in SHH binding to wildtype and Ptch1-/- MEFs.

Of course, overexpression studies have to be well-controlled and we provide several types of controls to ensure that the effects we see are physiologically relevant. Interfering with PTCH1 biochemical function using three completely different (but well-established) strategies-- a classical point mutation (D513Y, Figure 5), addition of its known inactivating ligand SHH (Figure 3) or dissipation of the K+ gradient (Figure 6)-- abolishes its effect on outer leaflet cholesterol. These controls exclude the possibility that the effects we observe are a non-specific artefact of overexpression.

In response to this comment, we have added a section to the *Discussion* titled “Limitations of this study” (lines 385-402) that thoroughly discusses the limitations of this assay, including the use of PTCH1 overexpression and the inability to selectively measure extraction of ciliary cholesterol. In addition, we conducted two experiments whose results are shown below:

To understand the degree of PTCH1 overexpression in our system, we used immunofluorescence to measure the abundance of PTCH1 at the plasma membrane of our HEK293T cells after Dox addition and compared it to the abundance of PTCH1 at primary cilia in a Hh-responsive, ciliated mouse embryonic fibroblast (MEF) cell line we have extensively used in prior publications to study the function of PTCH1 (Rohatgi et al., 2007). As shown in Author response image 1, the abundance of PTCH1 at primary cilia is comparable to the abundance of PTCH1 at the plasma membrane in the HEK293T cells used in our manuscript. Thus, the abundance of PTCH1 in the HEK293T cells used in our current manuscript is comparable to other systems where PTCH1 function, localization and signaling have been studied.

**Author response image 1. sa2fig1:** Anti-PTCH1 antibodies were used to measure the fluorescence of PTCH1 at primary cilia in mouse embryonic fibroblasts (MEFs) stably expressing PTCH1 (left) or at the plasma membrane of HEK293T cells used in our manuscript after Dox treatment (16 hours, 1 micromolar). Each dot represents one cilium (MEFs) or one cell (HEK293T cells). To show that the anti-PTCH1 IF signal is specific, PTCH1 measurements were also performed in MEFs treated with SHH (100 nM) for 16 hours (which is known to clear PTCH1 from cilia) and in HEK293T cells without Dox exposure.

As we suggested in the revision plan, we exposed cells to lower concentrations of Dox to see if PTCH1 could be induced to lower levels (Author response image 2) . As shown by the immunoblot (top), the dynamic range is limited, PTCH1 abundance rapidly increases between 0.1 nM and 10 nM of Dox. We conducted TIRFM assays to measure outer leaflet cholesterol accessibility at various concentrations of Dox (0.1-1 nM), with the results shown in (Author response image 2) (bottom). As we increase the Dox concentration from 0.1 to 0.3 to 0.9 nM, we observed a progressive decrease in outer leaflet accessible cholesterol-- shown by both a decrease in the initial extraction rate and a decrease in the maximum amount of cholesterol extracted. Thus, within the limitations of our Dox-inducible system, the influence of PTCH1 on outer leaflet cholesterol accessibility is dose-dependent and the effects are observable at significantly lower abundances of PTCH1. Additionally, new data in the revised manuscript (Figure 3G) shows that the effect of SHH (an inactivating ligand for PTCH1) on outer leaflet cholesterol is dose-dependent.

**Author response image 2. sa2fig2:** Time course of the change in TdTomato-PFOD4H membrane fluorescence after MβCD addition to HEK293T cells treated with increasing concentrations of Dox to induce the expression of progressively greater amounts of PTCH1. The immunoblot (top) shows whole cell PTCH1 abundance after treatment with increasing concentrations of Dox.

Reviewer #1 (Recommendations for the authors):[…]I don't disagree with the data interpretations but am having a hard time visualizing the biology of the system when PTCH is expressed because of the way the MBCD results are presented. The gist is that there is less free cholesterol in the outer membrane when PTCH is around, so MBCD doesn't have much to extract and PFO TIRF is not redistributing. However, since the lines don't shift following MBCD (Figure 2C as an example), it looks like cholesterol is locked in the membrane by PTCH, rather than already having been depleted. I think this is a visualization problem because everything is normalized to 1. Are the baseline readings significantly different between the samples? Is baseline higher/lower when PTCH is expressed? I needed to sketch it out for myself, and I feel like this is too much work for the reader. Is there a way to convey the results so they're more intuitive? More cartoons, maybe? The cartoons provided are very helpful.

We apologize that we did not clearly present the rationale for the way in which the kinetics of outer leaflet cholesterol extraction by MβCD were analyzed and presented. Briefly, our goal in establishing the TIRFM assay was to measure cholesterol accessibility in the membrane outer leaflet without relying on steady-state binding of PFO- or ALO-based probes. The steady-state binding of such probes has been used by us (in our 2019 *eLife* paper) and others (Zhang et al., 2018), with conflicting results (see the first section of the *Results*). The source of the conflicting results may lie in the high level of non-specific protein and membrane binding of the PFOD4 variants used in the papers from other groups. These issues have been extensively discussed in a previous *Scientific Correspondence* published in *eLife* (Courtney et al., 2018), where it was concluded that accurately inferring cholesterol content in membrane leaflets using hydrophobic PFOD4 variants used in (Zhang et al., 2018) should be accompanied by caution. Most importantly, the probes used by these other groups are not selective for accessible or active cholesterol (see Supplementary Figure 1e-k in (Liu et al., 2017)) over total cholesterol and so cannot be used to measure accessible cholesterol changes in response to Hedgehog signaling.

Given the pitfalls of using steady-state probe binding as the sole measure of accessible cholesterol, we developed a time-resolved, kinetic assay to monitor the rate of outer leaflet cholesterol extraction by MβCD. This assay is based on a decade of both experimental and theoretical work by many labs that are considered pioneers in membrane biology: McConnell, Lange, Steck, Slotte and others (Lange et al., 2004; Litz et al., 2016; McConnell and Radhakrishnan, 2003; Ohvo and Slotte, 1996; Radhakrishnan and McConnell, 2000). Indeed, careful measurement of the rates of cholesterol extraction by MβCD (not steady-state measurements of cholesterol content) first led to discovery of accessible cholesterol (summarized in (Lange and Steck, 2020)).

Kinetic assays are generally both more sensitive and less susceptible to artefacts. For example, the steady-state binding of PFOD4 to the inner leaflet is influenced by many factors other than the cholesterol content, such as the amount of PFOD4 expressed in an individual cell (noted by Reviewer #2) and the non-specific affinity of the probe for membranes. In contrast, cholesterol extraction by MβCD follows simple first-order, exponential decay kinetics: the half-life (t_1/2_) for extraction is independent of the starting baseline value (as it is for radioactive decay). In all the data presented in our paper, the curves represent the combined results from >20 individual cells (not single cells). Each of these cells shows a different baseline level of PFOD4H fluorescence at the membrane, likely because each cell expresses different amounts of the probe. This wide distribution in baseline values (Figure 1B) reduces sensitivity (e.g. while the median baseline values are different with and without PTCH1, the distributions overlap). We felt that the best way to analyze the data was to focus on the kinetics (or half-life of extraction) and to normalize the fluorescence for each cell to its own baseline value (set to 1) before averaging over all the cells analyzed. A classical paper on measuring accessible cholesterol in intact cells by Rothblat and colleagues (on which our assay is based) used this same analysis strategy (see Figures 1-4 in (Yancey et al., 1996)).

To show the impact of this normalization procedure, we now show data from our TIRFM assay analyzed in two different ways for select panels in Figures 2 and 3. In addition to baseline normalized data (Figures 2E-2F and Figures 3C, 3I), we provide panels (Figure 2—figure supplement 1C-1D and Figure 3—figure supplement 2A-2B) where the data is presented without any normalization to the baseline, with the raw fluorescence values (corrected for photobleaching) simply averaged at each time point over the ~20 cells analyzed. As this reviewer predicts and as shown by Figure 1B of the manuscript, the baseline values are indeed different: PTCH1 expressing cells start out with a higher level of PFOD4H binding to the inner leaflet. While the main conclusions from either representation are the same, it is much more difficult to compare the exponential decay phases (which are the focus of this assay) when the baselines are different. This is the reason we chose to represent the data as “fractional fluorescence remaining” relative to a value of “1” at baseline (Fluorescence at time t divided by Fluorescence at time zero).

Finally, we completely understand the statement that cholesterol looks “frozen” when PTCH1 is expressed. To address this issue we performed a dose-response curve using a range of MβCD concentrations in Figures 3D-3F of the manuscript. This experiment shows that cholesterol can still be extracted from PTCH1 containing membranes at higher concentrations of MβCD.

Re: Figure 6 and related to the point above: If PTCH is flipping cholesterol to the inner membrane, this should be reflected by PFO TIRF even in the absence of MBCD. Do you see a higher baseline in response to Dox induction of PTCH expression? The results presented in Figure 6 are less convincing that the rest of the figures, so it felt like the manuscript was ending with an afterthought. This point would be improved by additional experimental support and more discussion. There is only one sentence discussing the results in Figures 6B and C.

Thank you for this very helpful comment, which has prompted us to reorganize of the paper to address both the comments from Reviewer #1. As described in detail above in our response to Comment 1.1, we agree that the measurements using steady-state binding of probes are less sensitive and have a smaller dynamic range than measurements using the half-life of cholesterol extraction by MβCD. As this reviewer comment perceptively notes, this point is strikingly demonstrated when comparing the data shown in Figure 1 (steady-state binding) and the kinetic data shown in the rest of the manuscript. This is the reason our manuscript uses these kinetic measurements (rather than steady-state probe binding) to assess cholesterol accessibility in the membrane outer leaflet. As shown in Figure 1B and Figure 3—figure supplement 2A, we indeed see a higher baseline PFOD4H fluorescence at the inner leaflet when PTCH1 is expressed.

Prompted by this constructive comment, we have inverted the organization of the paper. We now begin by showing data on the effect of PTCH1 on the steady-state binding of TdTomato-PFOD4H or GFP-GRAM_1b_ to the inner membrane leaflet (Figures 1B and 1C) and mNeon-ALOD4 to the outer leaflet (Figure 1—figure supplement 1A). Once the reader is exposed to changes in baseline fluorescence caused by PTCH1, we then introduce the rationale for the kinetic assay used in the rest of the manuscript. We carefully describe the limitations of the steady-state binding assays and also explain the normalization process for the kinetic assays (lines 125-138). In addition to baseline normalized data (Figures 2E-2F and Figures 3C, 3I), we provide panels (Figure 2—figure supplement 1C-1D and Figure 3—figure supplement 2A-2B) where the data is presented without any normalization to the baseline, with the raw fluorescence values (corrected for photobleaching) simply averaged at each time point over the ~20 cells analyzed.

We thank the reviewer for this comment as the reorganized paper is more logical in its progression.

Reviewer #2 (Recommendations for the authors):Recent publications suggest that PTCH1 is a cholesterol transporter that mobilises cholesterol or a cholesterol derivative from the inner leaflet of the plasma membrane to the outer leaflet, where its concentration is about 10-fold higher (Zhang et al., 2018, Cell 175, 1352-1364). This transport against a concentration gradient requires energy, and previous literature indicates that sodium is the cation that provides such energy (Myers et al., Proc Natl Acad Sci U S A 2017;114(52):E11141-E11150).Related question – Conceptual concern: why would PTCH1 require energy provided by a cation gradient to move cholesterol in favour of its concentration gradient? The related cholesterol transporter NPC1 transports cholesterol from the outer to inner leaflet without an energy gradient, and it lacks the acidic triad essential for PTCH1 activity.

The mention of a cholesterol gradient between the outer and inner leaflets is raised several times in this review, including the related statement that “the inner leaflet cholesterol is 1%-3%.” The question of the transbilayer distribution of cholesterol in the plasma membrane is a controversial and unresolved issue. Work from multiple groups using diverse approaches supports a roughly equivalent (50-50) distribution of cholesterol in the two leaflets (these studies are nicely summarized in Table 1 of (Steck and Lange, 2018)). Moreover, these studies also established that the concentration of cholesterol in the plasma membrane is ~30-40 mole%.

In contrast, work from one group using PFO-based probes modified to markedly increase their hydrophobicity supports the notion that the concentration of cholesterol in the outer leaflet is 10-fold higher than the inner leaflet (Buwaneka et al., 2021; Liu et al., 2017). It is worth noting that two published commentaries by independent experts in the field (one published in *eLife*) have challenged these conclusions (Courtney et al., 2018; Steck and Lange, 2018). Importantly, the *eLife* paper provided experimental evidence that the PFO-based probes used by the Cho group (and which were also used by the papers that our work is compared to in these reviews) cannot be used to reliably infer cholesterol abundances in the two leaflets because of non-specific membrane and protein binding. Courtney and colleagues also show that the inner leaflet cannot contain only 1%-3% cholesterol because a PFOD4 based probe that binds membranes only above a cholesterol mol% of >35% readily binds to the inner leaflet (Courtney et al., 2018). In addition, PFOD4H binds to membranes at cholesterol concentrations greater than 30%, not 1-3% (Johnson et al., 2012). Yet, we (Figure 2C in the manuscript) and others (Abe and Kobayashi, 2021; Maekawa and Fairn, 2015) readily detect its binding to the inner leaflet. As we note in lines 368-382 of the *Discussion* our work does not depend on (and does not inform) the current unresolved debates around the transbilayer distribution of cholesterol. However, given the uncertainty around the transbilayer distribution of cholesterol, we cannot make any statements about whether PTCH1 is moving cholesterol up or down its concentration gradient.

Second, we note that the use of sodium as the cation that provides energy is also uncertain. In contrast to the work implicating a sodium gradient referenced by this reviewer (Myers et al., 2017), work from the Salic lab has implicated the potassium gradient in driving PTCH1 function (Petrov et al., 2020). Our direct assays for PTCH1 activity support a role for the potassium gradient, in agreement with the Salic work. It is reassuring that both our work and the Salic work converged on a potassium gradient despite using completely different assays for PTCH1 activity.

PTCH1 contains an acidic triad (EED), conserved in bacterial RND permeases, that is essential for its activity as Smoothened repressor and likely allows co-transport of the cation.In this study, Kinnebrew et al., follow their previous finding that indicates that sphingomyelin (SM) depletion potentiates Hedgehog signalling (Kinnebrew et al., eLife. 2019 Oct 30;8:e50051). SM localises in the outer leaflet of the plasma membrane and forms a complex with cholesterol, reducing availability of "free" or chemically-active cholesterol. Given that this suggests that increases in cholesterol concentration in the outer leaflet drives Smoothened activation, as opposed to the findings of the Beachy Lab, they re-evaluate PTCH1's transport directionality in the current study using total internal reflection fluorescence microscopy (TIRFM) to image reduction of binding of a cholesterol sensor domain fused to a fluorescent protein (TdTomato) to the inner leaflet when cholesterol is extracted from the outer lamella using methyl-β-cyclodextrine. The findings support the author's conclusion that PTCH1 reduces accessible outer leaflet cholesterol, but several methodological and conceptual questions remain:– MβCD pinches accessible cholesterol out of the membrane, but it is unclear if the rate of extraction is comparable to the unknown rate of transport by PTCH1, or if a rate-limiting aspect of the model confounds the interpretation of the rate of PFOD4H-TdTomato fluorescence reduction. Therefore, while the assay can be optimal to capture spontaneous flip-flopping, it may not be a faithful readout of a facilitated transport event.

We agree that the rate of cholesterol extraction by MβCD cannot be used as a direct measure of the rate of cholesterol transport by PTCH1 (and are careful not to state this anywhere in the manuscript). What we can say is that the accessibility of outer leaflet cholesterol in membranes expressing PTCH1 is lower based on MβCD extraction rates. This rate can still be used to infer PTCH1 activity because is affected by the inactivation of PTCH1 using three independent strategies: a classical point mutation (D513Y, Figure 5), addition of its known inactivating ligand SHH (Figure 3) or dissipation of the K^+^ gradient (Figure 6). Finally, we have used two positive controls (Figures 2E and 2F) to show that the extraction rate is sensitive to the abundance of accessible cholesterol in the membrane. In conclusion, while the rate of cholesterol extraction by MβCD cannot be used to measure the rate of cholesterol transport by PTCH1, it can be used as a measure of accessible cholesterol in the outer leaflet (the way it is used in the manuscript).

– PFOD4H-TdTomato fluorescence in the steady-state is more likely to represent the inner leaflet cholesterol content. Because the TIRFM signals are normalised to t=0, this crucial information is not available in most experiments other than in Figure 6. This would be particularly useful in cells expressing PTCH1 and PTCH1-DL2, and after a few minutes of SHH addition, as the affinity of PFOD4H for cholesterol is close to the reported inner leaflet cholesterol level (~ 1-3 mol%).

This point has been discussed thoroughly in our response to comments 1.1, 1.2 and 2.2 above. We have outlined the shortcomings of using steady-state binding of PFO-based probes (especially those used in the prior papers from other groups) to infer cholesterol content in each leaflet and highlighted that our use of MβCD extraction kinetics represents a completely orthogonal measure of outer leaflet cholesterol accessibility. Importantly, our kinetic assay measures cholesterol accessibility in the outer leaflet (where sphingomyelin is located), not the inner leaflet.

As noted in our response to comment 2.1 above, the inner leaflet is unlikely to contain 1-3 mol% cholesterol. PFOD4H binds to membranes at cholesterol concentrations greater than 30%, not 1%-3% (Johnson et al., 2012). Yet we (Figure 2C in the manuscript) and others (Abe and Kobayashi, 2021; Maekawa and Fairn, 2015) readily detect its binding to the inner leaflet.

In response to this request, we now show data from our TIRFM assay normalized in two different ways for select panels in Figures 2 and 3. In addition to baseline normalized data (Figures 2E-2F and Figures 3C, 3I), we provide panels (Figure 2—figure supplement 1C-1D and Figure 3—figure supplement 2A-2B) where the data is presented without any normalization to the baseline, with the raw fluorescence values (corrected for photobleaching) simply averaged at each time point over the ~20 cells analyzed.

– If myriocin treatment increases accessible cholesterol in the outer leaflet, wouldn't it increase the inner leaflet cholesterol content by flip-flop? If that is the case, the non-normalised TIRF signal should be higher at t=0 than in vehicle-treated cells.

Yes, as we show in Figure 2-figure supplement 1B, myriocin treatment leads to increased steady-state binding of the PFOD4 probe to the inner leaflet of the plasma membrane.

– The concentration of SHH ligand used to module cholesterol availability seems excessively high (10-6 M). The same group used 25 nM in the previous study as a "high, saturating SHH concentration", in line with most groups using SHH in the range of 50-100 nM. A dose-response effect will also inform if the IC50 of SHH in the TIRF assay agrees with its Kd for PTCH1 and the EC50 for stimulation of Gli-dependent transcription.– The lipid modifications of SHH play a key role in the asymmetric binding mode to PTCH1 dimers. The article does not detail the modifications of the SHH used.– Induction of PTCH1 expression increases absolute TIRF signals in Figure 6, suggesting higher inner leaflet cholesterol. However, this interpretation depends on equal total fluorescence, i.e. equal expression of the fluorescent sensor proteins regardless of PTCH1 expression and SHH signalling.– If PTCH1 reduces accessible cholesterol in the outer lamella, one would expect to observe a reduction in binding of an extracellular cholesterol sensor, equivalent to the PFOD4H-TdTomato.– The biggest limitation of this study is the reliance on overexpressed PTCH1. The effect of SHH addition to Hh-competent cells expressing endogenous PTCH1 (such as NIH 3T3 cells) on cholesterol sensor measurements is essential, even if it cannot be determined in the primary cilium membrane.– This study clashes with previous reports using confocal microscopy vs TIRF. One question that comes to mind is if a different conclusion would be drawn using the PFOD4H-TdTOmato sensor in standard confocal imaging to image changes in lateral membranes that are more readily exposed to solvent and SHH than the basal membrane attached to the substratum.However, I identified some areas that need to be addressed. I would like you to consider providing additional evidence for some of my key concerns:1 – Provide baseline absolute levels of fluorescence in each condition.

This point has been discussed thoroughly in our response to comments 1.1 and 1.2 above. We again note that the purpose of our assay is to measure the outer leaflet accessibility of cholesterol without using steady-state binding of PFO-based probes, which has been done before and yielded conflicting results (Kinnebrew et al., 2019; Zhang et al., 2018). In addition, the half-life of MβCD-mediated cholesterol extraction (the parameter used to infer cholesterol accessibility in our manuscript) is independent of the starting baseline fluorescence value given the simple exponential decay kinetics of the extraction reaction.

In response to this request, we now show data from our TIRFM assay normalized in two different ways for select panels in Figures 2 and 3. In addition to baseline normalized data (Figures 2E-2F and Figures 3C, 3I), we provide panels (Figure 2—figure supplement 1C-1D and Figure 3—figure supplement 2A-2B) where the data is presented without any normalization to the baseline, with the raw fluorescence values (corrected for photobleaching) simply averaged at each time point over the ~20 cells analyzed.

2 – Demonstrate that induction of PTCH1 does not affect total expression of PFOD4H-TdTomato.

A data panel showing that TdTomato-PFOD4H abundance does not change when PTCH1 is induced with Dox is provided in Figure 3—figure supplement 1A.

As we discuss in the response to Comment 1.1 and in the *Results section* of the revised manuscript (lines 125-128), this effect is relevant only for the steady-state measurements shown in Figure 1. The half-life of cholesterol extraction by MβCD follows exponential decay kinetics and is hence independent of the starting baseline fluorescence value. This feature is a major strength of our kinetic assay since it allows measurement of outer leaflet cholesterol accessibility in a manner independent of probe expression in cells. Finally, the reversibility of the PTCH1 effect with acute SHH exposure makes it unlikely that our results are due to changes in PFOD4H expression.

3 – Investigate the effect of adding more physiological concentrations of Shh (in the 10-50 nM range) to Hh-competent cells expressing the cholesterol sensor. Potential controls of acute silencing of PTCH1.

We used a higher concentration of SHH in this manuscript because we are working in a system in which PTCH1 is overexpressed. As a control to ensure that SHH at this concentration does not have off-target effects, we used (Figure 3) a widely-studied mutant of PTCH1 that cannot bind to SHH (PTCH1-ᐃL2). In animals ranging from flies to humans PTCH1-ᐃL2 fails to bind or respond to SHH but retains its ability to suppress SMO activity (Briscoe et al., 2001). SHH (even at the higher concentration used in our assays) has no effect on cells expressing PTCH1-ᐃL2, showing that its effects in our system are specific.

There is considerable evidence (both from our experience and from the published literature) that increasing PTCH1 expression increases the amount of SHH that must be added to overcome the inhibitory effect of PTCH1 on Hh signaling.

In response to this comment we have provided a SHH dose-response curve for our TIRFM assay in Figure 3G. The EC50 (concentration of SHH that causes a half-maximum increase in cholesterol accessibility) is ~160 nanomolar.

With respect to PTCH1, we are already using a Dox-inducible system so that in the same cell line we can assess outer leaflet cholesterol accessibility in the absence (-Dox) or presence (+Dox) of PTCH1. In addition, we inactivate PTCH1 acutely by adding SHH or dissipating the K^+^ gradient. Silencing PTCH1 by withdrawing Dox is possible but will not be acute due to the long half-life of the protein and latency of Dox washout.

4 – Testing changes in binding of an extracellular cholesterol sensor in the system.

We have attempted this experiment using recombinant PFOD4-GFP or ALOD4-GFP added to the extracellular medium. However, we cannot use an extracellular probe to monitor the kinetics of cholesterol extraction by extracellular MβCD using TIRFM. The >50 kDa PFOD4-GFP and ALOD4-GFP probes have poor and nonuniform access to the space between the cell and the coverslip, precluding the use of TIRFM, which is crucial for our time-resolved, kinetic assay (see *Discussion* under comment 2.10 below).

However, we have used steady-state binding of extracellular PFO* in our 2019 *eLife* paper to show that SHH causes an increase in cholesterol accessibility in the ciliary membrane of NIH/3T3 cells (a result in complete agreement with the conclusions of our current manuscript). As requested in this comment, we used flow cytometry to measure the binding of an extracellular cholesterol sensor (mNeonGreen-ALOD4) to the outer leaflet of the PTCH1 expressing HEK293T cells used for the kinetic studies throughout this manuscript (Figure 1—figure supplement 1). As with other steady-state measurements, the changes are small; however, they are consistent and support the model that PTCH1 expression (in a SHH-reversible manner) decreases outer leaflet cholesterol accessibility. Differences in the FACS-based binding assay may also be dampened because it requires detachment of cells from the tissue culture plate and solution staining prior to flow cytometry. Again, we emphasize that the kinetic assays based on MβCD extraction reveal much more significant differences spread over a larger dynamic range compared to steady-state binding measurements.

5 – Indicate source and molecular details of the SHH ligand used: is it lipidated, is the Ile-Ile mimic of the N-terminal palmitate? Please add this to the methods.If Shh is unmodified, could you test if dual lipidated SHH has the same effects at a lower concentration?

We apologize for not making this more clear in the methods. We are indeed using SHH carrying a double Ile at its N-terminus (known as SHH-C24II) to mimic the palmitate attached to native SHH. This protein has been widely used in the literature, including by us in our 2019 *eLife* paper and in all other papers on Hedgehog signaling we have published over the last 7 years (Kinnebrew et al., 2019). Pepinsky and colleagues first developed the SHH-C24II variant in a seminal paper over two decades ago and validated its use as an easy to purify and well-behaved substitute for the endogenous ligand (Taylor et al., 2001). Full details of the SHHC24II used in our manuscript are now provided in the Methods (lines 638-645).

– The contradictory findings are unlikely to be explained simply by a difference in the cholesterol sensor used (PFO-D4H-tdTOmato vs PFO-D4H tagged with a small solvatochromic fluorophore). Additional controls and a deeper discussion of the geometry of the assay and the potential impact of measuring MbetaCD-induces changes in fluorescence vs steady-state levels will be necessary to understand the differences.Also: 6 – I'd also like to see a deeper discussion on the potential explanations of the Zhang et al., results. You are creating a lively controversy in the field and would strongly benefit from trying to figure out why your conclusions are correct and the other is not when the systems are so similar in most ways.

The paper from the other group referred to in this comment did not use PFOD4H as a probe of inner leaflet cholesterol, but rather a triple mutant of PFOD4 (Y415A/D434W/A463W) coupled to a solvatochromic fluorophore which was produced as a recombinant protein and microinjected into cells (Liu et al., 2017; Zhang et al., 2020, 2018). This probe is considerably more hydrophobic than PFOD4H. An independent *ELife* paper has highlighted the flaws in these probes-- they show a high level of non-specific protein and membrane binding and hence do not accurately reflect membrane cholesterol content (Courtney et al., 2018). In addition, the authors themselves have shown that these probes cannot be used to measure accessible cholesterol in membranes (Supplementary Figure 1ek of (Liu et al., 2017)) and hence cannot be used to comment on changes in accessible cholesterol in response to SHH.

The assays used by our current manuscript and the work from other groups described in this comment are fundamentally different. Work from others used the steady-state inner leaflet binding of an highly engineered PFOD4H variant that does not selectively detect accessible cholesterol (shown in their own manuscript (Liu et al., 2017)) and hence has lost the critical property that makes it useful as a probe of how SHH/PTCH1 impact accessible cholesterol. In contrast, we have developed a time-resolved, kinetic assay to monitor the rate of outer leaflet cholesterol extraction by MβCD, a sensitive indicator of accessible cholesterol in cells (Haynes et al., 2000; Yancey et al., 1996). Our measurement of accessible cholesterol does not depend on steady-state probe binding (which has produced conflicting results), but rather on measuring the rate of cholesterol transfer to MβCD, an assay which has no similarity to the one used by other groups.

In response to this comment, we have significantly expanded our discussion of the differences between our manuscript and the work from others outlined above. We begin the Results with a section entitled *Conflicting models for the influence of PTCH1 on membrane cholesterol organization* (lines 43-93). In addition, we have also added a section in the Discussion section entitled *Comparison to alternative models for PTCH1 transporter function* (lines 331-353). We appreciate this suggestion as it provides a better context for our manuscript relative to the major competing models for the function of PTCH1 in cholesterol transport.